# Thermal legacy of a large paleolake in Taylor Valley, East Antarctica as evidenced by an airborne electromagnetic survey

Krista F. Myers[1], Peter T. Doran[1], Slawek M. Tulaczyk[2], Neil T. Foley[2], Thue S. Bording[3], Esben Auken[3], Hilary A. Dugan[4], Jill A. Mikucki[5], Nikolaj Foged[3], Denys Grombacher[3], and Ross A. Virginia[6]

[1]Department of Geology and Geophysics, Louisiana State University, Baton Rouge, LA 70803, USA
[2]Department of Earth and Planetary Sciences, University of California Santa Cruz, Santa Cruz, CA 95064, USA
[3]Department of Geoscience, Aarhus University, Aarhus, Denmark
[4]Center for Limnology, University of Wisconsin-Madison, Madison, WI 53706, USA
[5]Department of Microbiology, University of Tennessee, Knoxville, Knoxville, TN 37996, USA
[6]Department of Environmental Studies, Dartmouth College, Hanover, NH 03755, USA

*Correspondence to*: Krista F. Myers (kristafmyers@gmail.com)

**Abstract.** Previous studies of the lakes of the McMurdo Dry Valleys have attempted to constrain lake level history, and results suggest the lakes have undergone hundreds of meters of lake level change within the last 20,000 years. Past studies have utilized the interpretation of geologic deposits, lake chemistry, and ice sheet history to deduce lake level history, however a substantial amount of disagreement remains between the findings, indicating a need for further investigation using new techniques. This study utilizes a regional airborne resistivity survey to provide novel insight into the paleohydrology of the region. Mean resistivity maps revealed an extensive brine beneath the Lake Fryxell basin which is interpreted as a legacy groundwater signal from higher lake levels in the past. Resistivity data suggests that active permafrost formation has been ongoing since the onset of lake drainage, and that as recently as 1,500 – 4,000 yr BP, lake levels were over 60 m higher than present. This coincides with a warmer than modern paleoclimate throughout the Holocene inferred by the nearby Taylor Dome ice core record. Our results indicate mid to late Holocene lake level high stands which runs counter to previous research finding a colder and drier era with little hydrologic activity throughout the last 5,000 years.

## 1 Introduction

Lakes provide a refuge to life in some of the more extreme environments on Earth, especially in places with strong seasonality such as polar regions. Lakes are a relatively stable source of liquid water which can determine the ecological viability of a system (Gooseff et al., 2017). The evolution of lake basins is important to survival of local ecosystems and interpreting the history of lake levels and water availability can provide important insights into long-term ecosystem dynamics. Despite extremely low temperatures and minimal precipitation, the McMurdo Dry Valleys (MDVs – Fig. 1) are the site of a series of closed basin lakes fed by alpine glacial and snow melt via ephemeral streams as well as by direct subsurface recharge (Toner et al., 2017; Lawrence et al., 2020). Annual lake levels in the MDVs have been recorded since the 1970s which provide a window into the dynamic climatic and geologic drivers of the MDV hydrologic system (Fountain et al., 2016). Lake levels have overall risen since the beginning of observational records starting in 1903 during Robert Scott's first expedition (Scott, 1905), however periods of lake level stagnation as well as lake level drop have also been recorded (Fountain et al., 2016). The fluctuation of the MDV lakes alters the biological and chemical exchange between surface waters, soils, and groundwater,

making it critical for understanding MDV connectivity and ecosystem evolution (Doran et al., 2002; Gooseff et al., 2017; Foley et al., 2019).

The drivers of MDV lake level change represent a complex combination of climate, ice sheet extent, and basin hypsometry. During the Last Glacial Maximum (LGM) roughly 28.5–12.8 ka before present (BP) Antarctica underwent a period of ice

sheet thickening and advance (Hall et al., 2015). Stable isotope records from Taylor Dome (located roughly 100 km west of the MDVs) indicate mean annual air temperatures ca. 4-9 °C lower than modern during the LGM (Steig et al., 2000). Following the LGM (12,000–2,000 yr BP) regional temperatures were up to 5 °C warmer than modern conditions (Fig. 2) (Steig et al., 2000, Monnin et al., 2004). During the LGM, ice from the East and West Antarctic ice sheets was grounded within the Ross Sea extending to the edge of the continental shelf, roughly 500 km further than the modern-day grounding line (Anderson et

al., 2014). Grounded ice from the Ross Ice Shelf flowed inland, spilled into the ice-free MDVs, and formed an ice dam that reached approximately 350 meters above sea level (masl) (Denton and Marchant, 2000). The grounded Ross Ice Shelf in the mouth of Taylor Valley (TV) would have allowed for lake levels to reach elevations that would otherwise not be possible (~300–350 masl), forming a massive glacially dammed lake known as Glacial Lake Washburn (GLW), which would have filled all of Taylor Valley (Fig. 3a). According to Hall and Denton (2000), the Ross Ice Shelf remained grounded at a maximum

equilibrium extent along the eastern side of the Transantarctic Mountains until roughly 13,067 yr BP (12,700 [14]C yr BP, calibrated using Stuiver et al., 2017).

The removal of the ice dam in the mouth of TV would have resulted in drainage of GLW, and subsequent lake levels would be limited by the 78-81 masl topographic constraint (outflow sill) at Coral Ridge (Fig. 3b). However, the exact date that the

ice dam was removed is still not completely resolved in the literature. Some studies suggest the grounded ice sheet was completely gone from the mouth of TV sometime between 7,800 to 6,000 yr BP (Cunningham et al., 1999; Hall et al., 2006; Hall et al., 2013; Anderson et al., 2016). However, grounded ice could have persisted in the mouth of TV after the Ross Ice Shelf retreated. Levy et al. (2017) used luminescence dating to suggest that large paleolakes in neighboring Garwood Valley persisted well after the Ross Ice Shelf retreated. The paleolake Howard (Garwood Valley) was at its maximum elevation until

4.26 +/- 0.72 ka because of stranded ice in the mouth of Garwood Valley that was a relic of the Ross Ice Shelf. Levy et al. (2017) makes the case that relic ice could have persisted in the mouths of various valleys well after the Ross Ice Shelf retreated in McMurdo Sound, so ice shelf retreat may not correlate exactly with the timing of GLW drainage. Other studies (Horsman, 2007; Arcone et al., 2008; Toner et al., 2013) suggested that GLW was much smaller, and only occupied western TV in the Lake Bonney basin.


Much of the past lake level chronology in TV has been established by dating paleodeltas, which form where streams enter the lake and deposit sediment. Previous studies have utilized radiocarbon dating of algal mats within these deltas (Hall and Denton, 2000) and optically stimulated luminescence (OSL) dating of quartz grains (Berger et al., 2013) to constrain the age of the

paleodeltas. Hall and Denton (2000) postulate that a large lake occupied the full TV, reaching an elevation of up to 350 masl,
between a period of 8,340 to 23,800 yr BP, suggesting lake levels dropped below modern levels by 8 ka BP (Fig. 4). Berger et al. (2013) determined that delta deposits are systematically younger than the $^{14}$C dates at comparable elevations by ~5,000 yr (Fig. 4), suggesting a lake level drop below sill level between 5 to 8 ka.

The disagreement of spatial distribution and timing of paleo lake levels in TV highlights the need to further constrain lake level history. This study provides a third method for estimating lake level history during the previously unconstrained mid to late Holocene (5 ka to present) using a novel application of electrical resistivity data to identify the subsurface thermal legacy of paleo lake levels in TV.

## 2 Site Description and Field Campaign

The MDVs are located within the Transantarctic Mountains and make up the largest ice-free region in Antarctica (Levy, 2013). Mean annual valley bottom temperatures in TV range from -14.7 °C to -23.0 °C (Obryk et al., 2020) and the region receives between 3–50 mm water equivalent of precipitation per year (Fountain et al., 2010). Lakes in the MDVs have 3–5 m thick perennial ice covers and vary in volume, chemistry, and biological activity (Lyons et al., 2000). During the austral summer, a short melt window occurs between December to February which accounts for most of the stream input to the lakes (Chinn, 1987). Open water moats form around the perimeter of the lakes during the summer, and water loss occurs due to sublimation of lake ice and evaporation of open water (Dugan et al., 2013; Obryk et al., 2017).

Lake Fryxell, located in TV, is one of the largest lakes in the MDVs at roughly 5.5 km long, 1.75 km wide and 22 m deep (Fig. 1). Lake level measurements show that from 1972 to 2020, Lake Fryxell has undergone ~2.61 m of total lake level rise as measured by manual lake level surveys, at an average rate of ~5.4 cm yr$^{-1}$ (Fig. 5) (Doran and Gooseff, 2020). The lake level record is characterized by high interannual variability in both lake level change (magnitude) and direction (rise versus fall) (Fountain et al., 2016).

### 2.1 Resistivity Surveys

SkyTEM, a time-domain airborne electromagnetic sensor system, was flown over TV in 2011 (see Fig. 6 for SkyTEM data collection flight lines). The measurement involves pulsing a strong current in a transmitter coil suspended beneath a helicopter. When these currents are turned off rapidly, a primary magnetic field propagates throughout the subsurface, which induces eddy currents in the subsurface at depth. The decay of these eddy currents in turn generates a secondary magnetic field, the strength and time-dependence of which is measured inductively by a receiver coil that is also suspended beneath the helicopter (Foley et al., 2016). The resulting data, which consists of time derivatives of the secondary magnetic field, is used to estimate the resistivity in the subsurface (Ward and Hohmann, 1988). Data collection occurs while the helicopter flies at speeds up to

approximately 100 km h$^{-1}$. The 2011 airborne electromagnetic survey in 2011 collected 560 km of resistivity data in eastern TV. Flight lines were approximately 500 m apart, and nodes (individual sounding points) were collected every ~25 m along each flight line (Fig. 6). For more details and in-depth discussion behind SkyTEM methodology and data collection, see Sørensen and Auken (2004), Mikucki et al. (2015), and Foley et al. (2016).

Specialized inversion software, called Aarhus Workbench, was used to process raw resistivity data and invert for layer models of subsurface resistivity (Foley et al., 2016). Mean resistivity maps, which illustrate the mean resistivity of the subsurface at fixed elevation intervals, were generated in Workbench using a spatially constrained ('quasi-3D') inversion of the data collected from Fryxell basin in 2011 (Viezzoli et al., 2008). These maps are valuable for illustrating lateral continuity and extent of resistivity structures at depth. Five-meter-thick slices of constant elevation from 250 masl to -300 masl were
interpolated to generate mean resistivity maps using a 1,000 m search radius and kriging interpolation. Kriging involves the use of a semi-variogram to determine weights during the interpolation, which makes this method well-suited to capturing spatial correlations. The variogram model used in the kriging interpolation is a simple exponential function with log-transformed resistivity values, a sill value of 0.16 and a range of 1,520 m.

Mean resistivity maps were imported into ArcGIS and then overlaid on digital elevation models (Fountain et al., 2017) and high-resolution Digital Globe satellite imagery (WorldView 3, provided by the Polar Geospatial Center) to visualize the location and extent of low resistivity regions. The digital elevation model used for this study was generated from a 2015 LiDAR campaign flown over the McMurdo Dry Valleys in 2015. The digital elevation model has 1 m spatial resolution and covers all of TV (Fountain et al., 2017).


The depth of investigation was determined in Workbench and represents the maximum depth of reliable resistivity models. Regions of low resistivity (<1,000 Ωm, typical of lake water and brine saturated sediments) have a lower depth of investigation (<100 m) due to signal attenuation in conductive materials, whereas regions of high resistivity (>1,000 Ωm, typical of permafrost and bedrock), have a greater depth of investigation (>300 m) because the EM signal penetrates deeper with less
attenuation in these conditions (Christiansen and Auken, 2012; Mikucki et al., 2015; Foley et al., 2016). We are using the threshold of ~1,000 Ωm to broadly distinguish between "high" versus "low" resistivity based on observations from both McGinnis and Jensen (1971) as well as the Mikucki et al. (2015), which broadly defines sediments as brine-bearing if they have resistivity from 10 to 1,000 Ωm.

Subsurface characteristics and chemistry can be inferred from the relationship of bulk resistivity to temperature, salinity, and liquid volume fraction (Mikucki et al., 2015). Permafrost resistivity varies, and can be broken into subgroups depending on degree of saturation and confining properties as described in McGinnis and Jensen (1971). Confining permafrost does not allow any fluid flow, whereas aquifrost is permafrost that allows groundwater flow due to local conditions such as salinity and

porosity (McGinnis and Jensen, 1971). Confining layer permafrost tends to have much higher electrical resistivities (<10,000

Ωm) than aquifrosts (50 - 1,000 Ωm) depending on temperature and degree of saturation of the aquifrost (McGinnis and Jensen, 1971).

The regions of low resistivity underlying higher resistivities are interpreted as liquid brine saturated sediments capped by permafrost and represent the liquid groundwater extent in Fryxell basin (Mikucki et al., 2015). To calculate the thickness of

the permafrost above this briny aquifer, we performed a search from the surface to the first instance at depth of resistivity values at or below a threshold value of 100 Ωm. The 100 Ωm threshold broadly distinguishes between "dry" or hard-frozen permafrost (above 100 Ωm) and aquifrost or unfrozen ground with a significant liquid fraction in pore spaces (Mikucki et al., 2015; Foley et al., 2016). We chose 100 Ωm as our cut off based off of Foley et al. (2016, Fig. 7) Dry Valley Drilling Project borehole data comparison that shows resistivity measurements. The borehole DVDP-11 was drilled within our study area and

temperature and salinity was measured within the borehole. These borehole data can be compared to our resistivity profile taken near the borehole site (Foley et al., 2016, Fig. 7). Foley et al. (2016) shows a rapid decline in resistivity from about 100 Ωm down to <5 Ωm within only 20-30 m. This sharp resistivity gradient corresponds well to an abrupt jump in salinity with depth. Such a salinity increase is consistent with a transition from frozen sediments to liquid brine.

Workbench was then used to export location, elevation, and depth to brine for each node point of resistivity data collected. Depth to brine values for each node were interpolated to map distribution of the permafrost/brine boundary. Depth to brine maps were smoothed using a low pass filter in ArcGIS to reduce noise, which was particularly high around the edges where data density was lower and signal-to-noise ratios were less favorable.

## 2.2 Modelling permafrost thickness and age, analytical model

Permafrost thickness, which is taken as the depth to 100 Ωm boundary, was used to calculate time since initiation of permafrost freeze-back resulting from lake drainage. We adapted Eq. (5) from Osterkamp and Burn (2003);

$$t = s\frac{d^2 \cdot H_s}{2k_b \cdot (-T_{ps})}$$ Eq. (1)

where $t$ is the age of permafrost in years, $s$ is a conversion factor from seconds to years, $d$ is the thickness of permafrost (m), $H_s$ is the volumetric latent heat of fusion for sediments corrected for porosity in Joules per cubic meter (Jm$^{-3}$), $k_b$ is the bulk thermal conductivity of permafrost in watts per meter per Kelvin (Wm$^{-1}$K$^{-1}$), and $T_{ps}$ is the temperature difference between the atmosphere and permafrost freezing front (K). The average air temperature of Lake Fryxell is -20 °C (Obryk et al., 2020), and freezing temperature of brine saturated sediments is between –5 °C to -10 °C (Foley et al., 2016). Here we use an average

brine freezing temperature of -5 °C, and a max brine freezing temperature of -8 °C. Therefore, the average $T_{ps}$ is -15 K and maximum $T_{ps}$ is -12 K. The volumetric latent heat of fusion for sediments corrected for porosity is estimated using Eq. (2);

$$H_s = H_v \cdot \varphi \cdot (1 - \varphi_a) \qquad\qquad \text{Eq. (2)}$$

where $H_s$ is the volumetric latent heat of the sediments, $H_v$ is the volumetric latent heat of water (334 MJ m$^{-3}$), $\phi$ is the porosity of the sediment, and $\phi_a$ is the fraction of air in pore space. Bulk thermal conductivity of permafrost ($k_b$) is estimated using Eq. (3);

$$k_b = k_m{}^{1-\varphi} \cdot k_f{}^{\varphi(1-\varphi_a)} \cdot k_a{}^{\varphi \cdot \varphi_a} \qquad\qquad \text{Eq. (3)}$$

where $k_m$ is the thermal conductivity of the matrix (Pringle, 2004), $k_f$ is the thermal conductivity of the fluid (Engineering Toolbox, 2004), and $k_a$ is the thermal conductivity of air (Engineering Toolbox, 2009). All thermal conductivity terms are in units of W m$^{-1}$ K$^{-1}$ and values are summarized in Table 1.

To calculate a range of possible permafrost ages for each elevation, a Monte Carlo statistical analysis was performed. Each input variable was assigned a standard deviation. $\phi$, $\phi_a$, $T_{ps}$, and $k_m$ were assigned a standard deviation of 20%, and $k_f$ and $k_a$ were assigned a standard deviation of 2% as they are better constrained. We then draw 10,000 realisations, where each input variable is drawn from a normal distribution with mean value and standard deviation as listed in Table 1. For each realisation we calculate permafrost age assuming completely saturated sediment ($\phi_a = 0$) as well as partially saturated sediment. This approach assumes a normal distribution of all parameters, and a homogeneous substrate in both space and time for simplification purposes.

To produce a deterministic upper bound of permafrost ages, Eq. (1) was calculated using the maximum or minimum possible value within one standard deviation for each input variable. $H_s$ was maximized by assigning a maximum $\phi$ (adding one standard deviation), with and without air in the pore space. Minimum $k_b$ was calculated by minimizing $k_m$ by subtracting one standard deviation and maximizing $\phi$ by adding one standard deviation, with and without air in pore space. $T_{ps}$ was minimized by adding one standard deviation from the average value (Table 1).

**2.3 Modelling permafrost thickness and age, 1D numerical vertical diffusion model**

The model in section 2.2 uses Eq. 1 to calculate permafrost age, assuming a constant $T_{ps}$ through time for simplification, however paleotemperature reconstructions point to a gradual cooling during the Holocene (Steig et al., 2000, Monnin et al.,

2004). In order to explore this further, we used a second approach to calculating permafrost ages using a 1D numerical (finite-difference) model solving the classical Stefan problem of vertical heat diffusion coupled with latent heat release during freezing. The upper boundary condition is a prescribed temperature that is $|T_{ps}|$ lower than the freezing point of the sub-permafrost brines. $T_{ps}$ can be either held constant during numerical experiments (like described in Section 2.2) or can be prescribed to vary with time. The bottom boundary condition of the numerical model is a constant heat flux, set to the geothermal flux of 0.080 Wm$^{-2}$ consistent with two borehole-based estimates proximal to the study area (boreholes DVDP-6 and CIROS-1 in Table 1, Morin et al., 2010). Other model parameters are based on the same permafrost properties listed in Table 1. Existing observational constraints indicate that under modern conditions in the study area, the temperature at the bottom of the permafrost is ca. -10 °C (e.g., Figure 7 in Foley et al., 2016) while ground surface temperature is ca. -20 °C (Obryk et al., 2020), yielding $T_{ps}$ of about -10 °C. We then applied a linear cooling rate of 1, 2, 3, and 4 °C over the last 10,000 yrs to model the cooling trend observed in the Holocene Taylor Dome paleotemperature reconstructions (Steig et al., 2000, Monnin et al., 2004). The ice core constraints appear to be best approximated by the linear cooling trend of 3 °C per 10,000 years. The numerically obtained permafrost thickness evolution with time can be fit well (R$^2$=0.9996) with an empirical power relation :

$$d = 4.52 * t^{0.424} \qquad \text{Eq. (4)}$$

where (like Eq. 1 above), t is the age of permafrost in years and d is the thickness of permafrost in meters. Note that Eq. (1) can be re-written in the same power-law form with a time exponent of 0.5 rather than 0.424.

## 3 Results

Airborne resistivity data were used to map groundwater and permafrost extent within Fryxell basin. A large low resistivity region (<100 Ωm) extends hundreds of meters away from the modern lake extent (Fig. 6, Fig. 7). The edges of the resistivity map shown in Fig. 6 commonly show a very strong gradient of resistivity, which are mostly artifacts of the interpolation. Defining the interpolation search radius (here we chose 1,000 m) is a balance in allowing enough overlap between surrounding model nodes to avoid gaps in the spatial mapping. However, a larger search radius does produce some interpolation artifacts around the edges. The interpolation artifacts are not a concern for regions where data coverage is good (for example, in the center of map, Fig. 6) and the search radius is interpolating between nodes on all sides.

In addition to mapping groundwater, higher resistivities overlaying low resistivity brine in the valley floor are interpreted as regions of permafrost which range from ~5 m thick around the lake edge to over 200 m thick higher up on the valley walls (Fig. 7). Permafrost is thinnest near the modern lake edge and increases in thickness with increasing distance from the lake edge (Fig. 8). Permafrost thickness frequency displays a bimodal distribution, with one mode <50 m thick (near lake edge)

and another peak around 150 to 200 m which occurred further up the valley walls (Fig. 9). Permafrost resistivity varies depending on water content and temperature. Confining permafrost generally had values above 10,000 Ωm, which was only observed in some regions of Fryxell basin (Mikucki et al., 2015, McGinnis and Jensen, 1971). A lower resistivity permafrost layer (between 100 to 1,000 Ωm) extends from the brine layer to approximately 81 masl (Fig. 7).

Calculated permafrost ages are plotted against ranges of possible porosity, bulk thermal conductivity, and temperature differences (Fig. 10). The effect of a partially saturated versus fully saturated substrate on the volumetric latent heat of fusion for sediments corrected for porosity ($H_s$) and bulk thermal conductivity ($k_b$) was also investigated (Fig. 10). We assumed partially saturated conditions to yield the oldest possible permafrost ages, however we recognize that assuming a homogeneous partially saturated substrate at this permafrost/brine boundary is an oversimplification.

Lower depth to brine values (thinner permafrost) found at low elevations (<80 masl) yielded a smaller range in possible ages. A wide range of calculated ages is seen at higher elevations (>80 masl), and therefore the deeper/older permafrost ages are less constrained (Fig. 11). Maximum permafrost ages for partially saturated conditions (0.24 air fraction in pore space) yielded slightly older ages compared to the first order calculation using average inputs and fully saturated conditions, resulting in a 245 ~500-year increase in age (approximately 1.5 to 1 ka BP) for shallow brines between "average" and "maximum" results (Fig. 11). Figure 12 shows the range of possible ages for a discrete depth to brine (permafrost thickness) of 50 m, 140 m, and 200 m as calculated from the Monte Carlo analysis for both partially saturated and fully saturated conditions. The age uncertainty is smallest for shallower brines (thinner permafrost) compared to deeper brines (thicker permafrost) (Fig. 12). The middle panel of Fig. 12 shows the age probability distribution for 140 m thick permafrost, which roughly corresponds to the 81 masl 250 sill level.

Permafrost age was mapped using both methods outlined in sections 2.2 and 2.3. First, we mapped permafrost age using the analytical model (using max value, partially saturated inputs in Table 1) which calculated permafrost age using depth to brine/permafrost boundary and Eq. 1 (Fig. 14). We then mapped permafrost age using the 1D numerical vertical diffusion 255 model using the best fit (Eq. 4) from Fig. 13 assuming a 3 °C linear cooling trend of the last 10 ka (Fig. 15). The age of permafrost was capped at 30 ka in Fig. 15 because this method is not as accurate for very thick permafrost (around the edges of the map), and so this model is best used for younger and thinner permafrost. Both methods produce relatively young permafrost ages around Lake Fryxell, however the model based on 1D numerical vertical diffusion output (Eq. 4) yields slightly older ages.

Permafrost ages from the two methods were then plotted as a function of sounding point elevation to compare to previously reported [14]C and OSL ages at corresponding elevations (Fig. 16, Fig. 17). A comparison of all three methods shows that even maximum permafrost age calculations cannot reproduce [14]C ages of paleodeltas from Hall and Denton (2000). OSL dates are

in rough agreement with permafrost age calculations, especially for low elevations surrounding the modern lake edge (<30 masl) (Fig. 16, Fig. 17).

## 4 Discussion

In polar regions, ubiquitous permafrost isolates surface water from deeper groundwater systems by forming a relatively impermeable layer. Exceptions occur where large lakes form and have formed in the past. This is because adding a lake and changing lake levels drastically alter the surface thermal boundary conditions from e.g. -20 °C (i.e., the typical mean annual air temperature in Taylor Valley - Obryk et al., 2020), when no lake is present, to around 0 °C when year-round liquid water occupies the surface. Airborne resistivity data revealed a low resistivity groundwater signal that extends hundreds of meters away from the modern lake edge of Lake Fryxell. In the MDV, this groundwater system is overlaid by permafrost 0 to 100s of meters thick. We propose that this region of low resistivity beneath higher resistivity permafrost is a relic thaw bulb from higher lake levels of the past. The presence of a large paleolake would have altered the thermal regime of inundated regions, allowing for a broader groundwater connection throughout much of TV. As the lake level dropped to modern levels, the paleo-lakebed was exposed to subfreezing surface temperatures resulting in permafrost growth from the top down. Gradual freeze-back of past lake beds following lake level drop (transient talik) has been observed in the Arctic, and permafrost continues to form until steady state conditions are reached (Neill et al., 2020).

Permafrost age calculations can be compared to paleotemperature records from Taylor Dome to correlate paleoclimate events with hydrologic responses. Lake levels appear to have been the highest around or after 12 ka BP (OSL date from ~250 masl), the end of a warming period from 15 to 12 ka BP (Steig et al., 2000). Lake levels likely remained high from 12 to 8 ka BP and subsequently dropped due to the retreat of the grounded Ross Ice Shelf from the mouth of TV (Hall and Denton, 2000, Hall et al., 2004, Anderson et al., 2016, Spector et al., 2017). After the removal of the ice dam, lake levels would not be able to exceed the sill level at Coral Ridge (Fig. 1). We measure sill elevation to be 78 – 81 masl based on the DEM of Fountain et al. (2017). Since channel erosion rates are unknown, we use the present-day bank elevation (81 masl) as the topographic threshold that defines the boundary of the Lake Fryxell watershed. Any additional hydrologic input into Lake Fryxell above this 81 masl level would spill into the McMurdo Sound via the channel that cuts through Coral Ridge (Fig. 1).

During the late Holocene, a much larger talik would have existed beneath a larger paleolake, and the thickness of permafrost above this relic talik can be used to calculate time since lake drainage. This study uses both an analytical model (Eq. 1) and a numerical model (Eq. 4) to calculate permafrost age. Neither of the two models take into account the rate of lake level drop, and both model results would be the same if we assume a constant rate of lake level drop or an instantaneous lake level drop. The analytical model assumes constant $T_{ps}$ for simplification, however we explored the scenario of a variable $T_{ps}$ (3 °C cooling over 10,000 years) in the numerical model (Eq. 4, Fig. 13, Fig. 15). Lateral heat flux from modern and ancient lakes is not

addressed in this study, however it may be important to understand equilibrium thresholds for this evolving groundwater system.

When it comes to analyzing the potential importance of putative lake level variability at elevations <81 masl, it is important to note in our Fig. 9 that the cluster of thin permafrost at low elevations occurs for elevations <50 masl where permafrost is estimated to be <50 m thick. Comparatively few of our estimates of permafrost thickness fall between 50 and 140 m with corresponding surface elevations between 50 and 81 masl. The only other high-abundance cluster of permafrost thickness is 140-200 m with corresponding surface elevations between 81 and 150 masl. This cluster represents the lake levels that occurred when the ice dam was still present. When the ice dam is not present, the hydrology of Fryxell basin favors lake levels that are considerably lower than the sill level of 81 masl and that reach mostly up to ca. 50 masl, which corresponds to the other cluster of common permafrost thicknesses that are <50 m thick.

Permafrost age calculations indicate late Holocene lake level high-stands (up to ~81 masl, 63 m higher than modern Lake Fryxell) roughly 1.5 ka BP (Fig. 14) to 4 ka BP (Fig. 15) that would have filled both Lake Fryxell and Lake Hoare basins (Fig. 3b). Permafrost ages span a wide range of elevations because the depth to brine boundary only roughly follows surface topography (Fig. 16, Fig. 17). This is probably an indication of subsurface heterogeneity (bedrock, till, lacustrine sediment, and buried ice distribution) which is also apparent in resistivity maps. Shallower depth to brine values (younger ages) are better constrained compared to older and higher elevation brines as suggested by Monte Carlo statistical analysis and can be used as an estimate of uncertainty (Fig. 11, Fig. 12). For this reason, we are not using this method to necessarily date the timing of the highest lake levels. This paper discusses GLW in the context of past lake level highstands, however other methods are likely better at determining very old lake levels compared to this method of using resistivity data.

Application of a linear cooling of 3 °C in the last 10,000 years to a numerical model of permafrost growth (Eq. 4) still yields fairly young ages for elevations below the 81 masl sill level (<150 m thick permafrost results in ~4ka, green line below). This would be about 2.5 ka older than our original approach using the analytical solution (Eq. 1) (Osterkamp and Burns, 2003). Even though the results vary between the two methods, the data still suggest that a larger lake occupied Fryxell basin between 1.5 ka (analytical method) to 4 ka (1D vertical diffusion model explained above). At higher elevations with permafrost thicknesses >/=200 m we obtain ages between 7 – 8 ka.

Using two methods to calculate permafrost ages from resistivity data, it was still not possible to yield ages for lake occupation that fully agreed with those estimated by [14]C ages of delta deposits (Hall and Denton, 2000) (Fig. 16, Fig. 17). Radiocarbon dating from Hall and Denton (2000) suggests that lake levels were higher than modern between 22 to 5 ka BP. OSL dates (Berger et al., 2013) estimate past lake level high stands to be more recent, existing between 12 – 5 ka BP. Neither OSL or [14]C suggest lake level high stands younger than 5 ka BP. Our permafrost age calculations agree more with the OSL chronology

than the [14]C at all elevations (Fig. 16, Fig. 17), but both chronologies have their limitations. OSL records the burial age of the quartz grain, which may be different from the deposition age. Rates of past sediment transport and deposition are not currently known, so this lag in deposition versus burial time is unresolved. Secondly, OSL dates are collected from some depth below the surface, meaning the dates may not be an accurate representation of the most recent occupation of that delta/lake level elevation (this will also be true for [14]C samples collected away from the surface). Several studies have shown a substantial

radiocarbon reservoir effect in the MDVs, but modern lake edges and streams have been shown to be mostly-well equilibrated with the modern atmosphere (e.g., Doran et al., 1999; Hendy and Hall, 2006). Doran et al. (2014) did note some exceptions. In fact, out of 4 moat microbial mat samples dated (all in < 1 m of water), only one was equilibrated with the modern atmosphere. The others carried [14]C ages of 2324±96, 9334±71, 2608±48. So clearly, even in modern lakes it is possible for lake edge material to carry a significant carbon reservoir. A large glacial lake of the past may have had even more ancient

unequilibrated carbon associated with it due to melt coming from the ancient Ross Ice Shelf without the opportunity for significant equalization with the atmosphere (e.g., through direct subaqueous glacial melt and moats being more closed to the atmosphere). Moreover, under climates colder than today lake ice may have been thicker and [14]C equilibration with ambient atmosphere more restricted. Levy et al. (2017) points out that [14]C have consistently produced dates 5 - 10 ka older than OSL samples from the same locations in Garwood Valley.

Multiple lines of evidence support our permafrost age calculation. Permafrost at approximately 81 masl dates at 4 ka and 1.5 ka BP using the numerical and analytical models, respectively. During this time, Taylor Dome ice core records show a highly variable Holocene climate with temperatures up to 5 °C above modern until 1-2 ka BP, when temperatures came down to modern levels (Steig et al., 2000, Monnin et al., 2004). Multiple studies in the MDVs make a direct connection with air

temperatures and hydrologic response (Obryk et al., 2017, Doran et al., 2008, Doran et al., 2001), which supports a potential connection between higher lake levels during the relatively warm Holocene. Also, at roughly the sill level (~81 masl), there is a large and very well-preserved delta on Crescent Stream in Fryxell basin, with nearly zero slope and sharp transitions between the topset and foreset suggesting low degrees of weathering (Fig. 18). Weathering of delta deposits has not been rigorously addressed in previous literature and we cannot quantify the relationship between delta appearance with age because there are

many unknown variables (i.e., deposition rate, geology, duration of lake occupation, and erosion rate); however, through visual comparison in the field as well as using DEMs and satellite imagery, the delta on Crescent Stream is the most well defined delta in Fryxell basin. Because this delta occurs at 81 masl, the maximum possible lake level without grounded ice at the mouth of TV, the delta may have formed through multiple cycles of filling and drainages. Lake Fryxell could have reached 81 masl multiple times in the past, and we suggest that periods with warmer than modern conditions such as the late Holocene could

result in this common lake level elevation, and any additional hydrologic input would result in spill over into the McMurdo Sound. The combination of a highly preserved delta deposit at 81 masl, the approximate permafrost age of 1.5 ka to 4 ka BP, and warmer than modern temperatures in the late Holocene, suggests that it was possible for lake level be at the sill level sometime in the last few thousand years. This is somewhat at odds with previous studies that have shown evidence for

drawdown events (significantly lower than modern) between 1-4 ka BP (Lyons et al. 1998), or at 6.4, 4.7, 3.8 and 1.6 ka BP (Whittaker et al, 2008). A partial desiccation event, or events, in the late Holocene cannot be resolved using either paleodelta age or resistivity data from this study.

## 5 Conclusions

Lake levels in TV have fluctuated in the past, leaving behind a complex history of overprinted lacustrine deposits and subsurface thermal signatures. A novel approach using airborne transient electromagnetic surveys provides new evidence that supports the existence of much higher lake levels in Fryxell basin by constraining lake level timing and extent. This study provides new insight on lake level evolution in TV during the Holocene, a period in which past studies are contradictory and significant gaps remain. Lake levels were higher potentially during and after the LGM when an ice dam blocked the mouth of TV, allowing for lake levels to increase by over 280 m compared to modern level. Taylor Dome ice core records indicate an abrupt warming of >15 °C from 15 – 12 ka BP, (Steig et al., 2000), which may have coincided with the maximum lake level of GLW. Following ice sheet retreat approximately 8 ka BP, GLW drained and lake level likely fluctuated from at or below modern levels (18 masl) up to 81 masl between 8 to 1.5 ka BP. Around 4 – 1.5 ka BP, lake levels were at the 81 masl sill level and have subsequently dropped to or below modern levels. Our chronology of Fryxell basin lake level history based on modeling of the thermal legacy of the lake is better supported by previously published OSL ages than [14]C ages of paleodeltas on the valley walls.

Small changes in climate such as a 5 °C higher temperatures in the late Holocene (Monnin et al., 2004) could have sustained anywhere between 60 to 80 m higher lake levels. Closed basin MDV lakes are characterized by high variability and extreme sensitivity to both climate and geologic drivers. Modern and paleohydrologic evidence indicates a highly dynamic system in which modest temperature forcings can initiate a large-scale hydrologic response.

## Data availability

Lake level data for Fig. 5 is available at https://mcm.lternet.edu/content/lake-level-surveys-mcmurdo-dry-valleys-antarctica-1991-present (Doran and Gooseff, 2020).
SkyTEM data collected in 2011 is available at https://www.usap-dc.org/view/dataset/601071

## Author Contributions

KMyers wrote the text, conducted data analysis, and made the figures. PDoran and STulaczyk helped write the text as well as develop methods and concepts. STulaczyk and NFoley provided conceptual framework for paper including the method to

determine permafrost age from Osterkamp and Burns (2003) and from numerical modeling. EAuken, TBording, NFoged, and DGrombacher provided technical details and support regarding electromagnetic data and processing, as well as edits to the text. TBording also provided help on Monte Carlo simulations. HDugan, JMikucki, and RVirginia all contributed to SkyTEM

data collection, conceptual framework for the project, and edits to the text and figures.

**Competing Interests**

The authors declare they have no conflict of interest.

**Acknowledgements**

We would like to thank the two reviewers as well as the editors of The Cryosphere for their excellent suggestions which greatly

improved the manuscript. This research is funded by the National Science Foundation (NSF) Grant #OPP-1637708 for Long Term Ecological Research. This material is based upon work supported by the NSF Graduate Research Fellowship under Grant No. 1247192. Support for KMyers and PDoran was also provided by the Louisiana State University John Franks Chair Fund. Contributions of STulaczyk and NFoley to this work were supported by two grants from NSF, 1344349 and 1644187. Contributions of JMikucki were supported by NSF grant number OPP-1344348. Contributions for RVirginia were supported

by NSF grant number 1043618. Other support for this project was provided by NSF grant numbers 1643687, 1643536, and 1643775. Geospatial support for this work provided by the Polar Geospatial Center under NSF-OPP awards 1043681 and 1559691. Any opinions, finding, conclusions, or recommendations expressed in the material are those of the authors and do not necessarily reflect the views of the National Science Foundation.

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

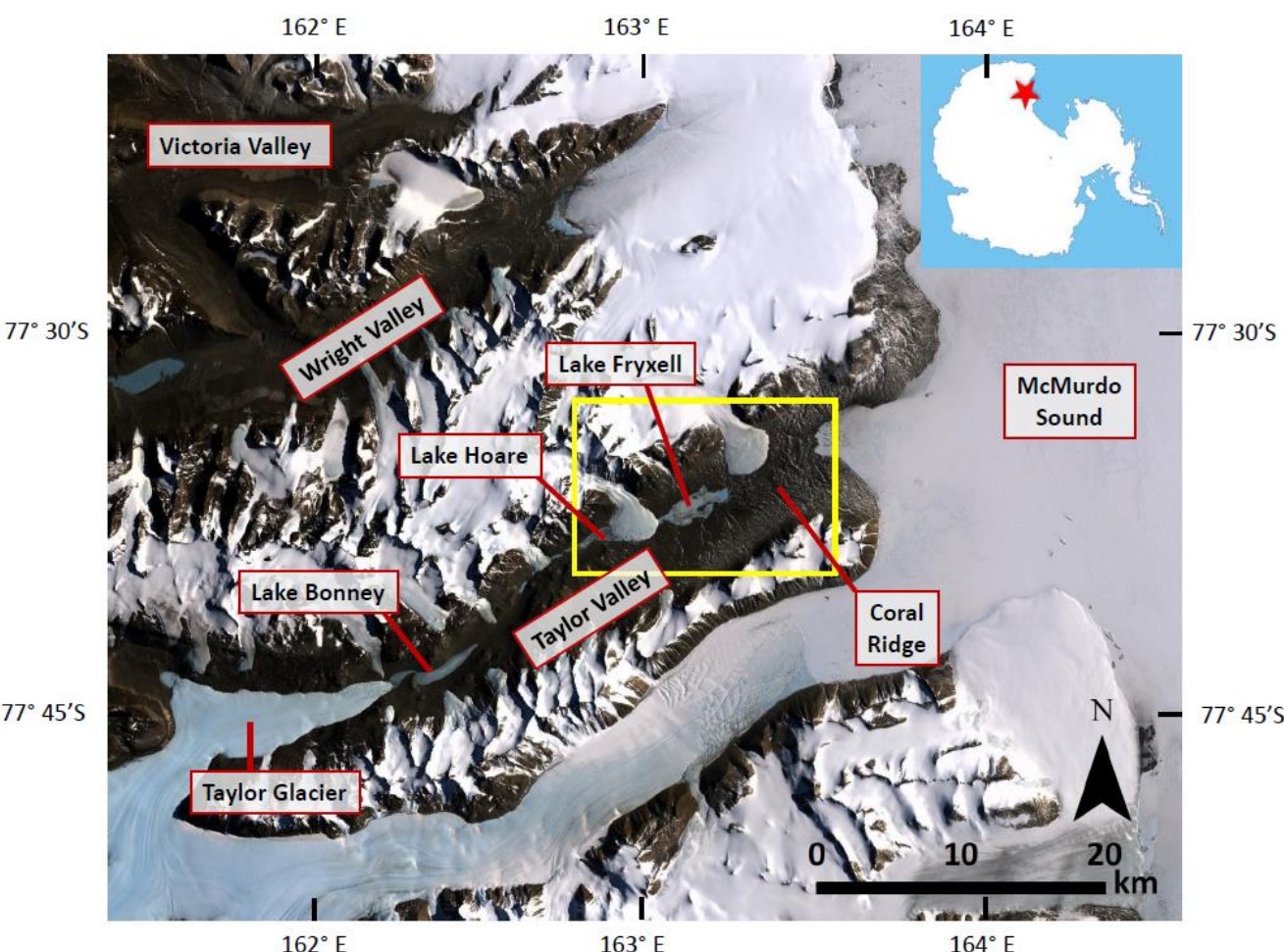

Fig. 1. Overview map of the McMurdo Dry Valleys. Lake Fryxell is located in the eastern portion of Taylor Valley, and the study area is outlined in yellow. Satellite imagery from Landsat Imagery Mosaic Antarctica (LIMA) published 2009.



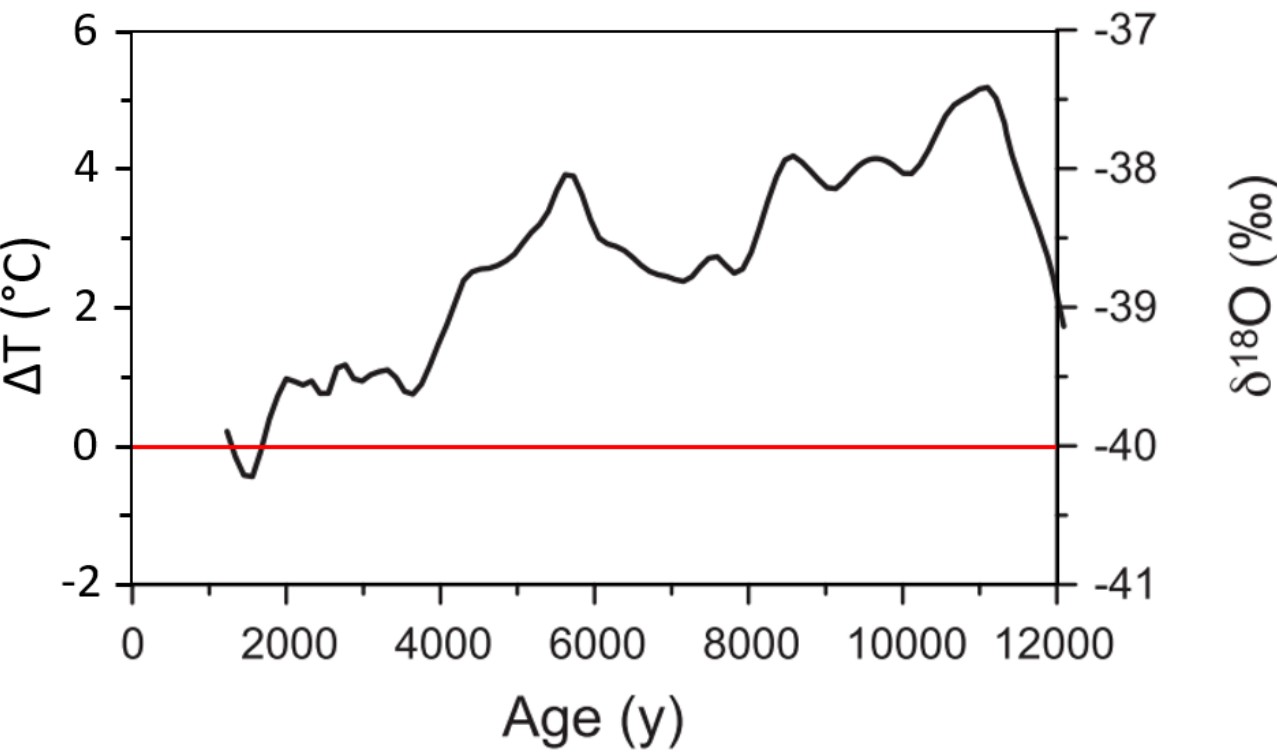

Fig. 2. Stable isotope (δ18O) record from Taylor Dome ice core and paleotemperature reconstruction. A reference line for 0
ΔT (°C) is shown in red, representing deviation from modern temperatures. Modified from Monnin et al. (2004)




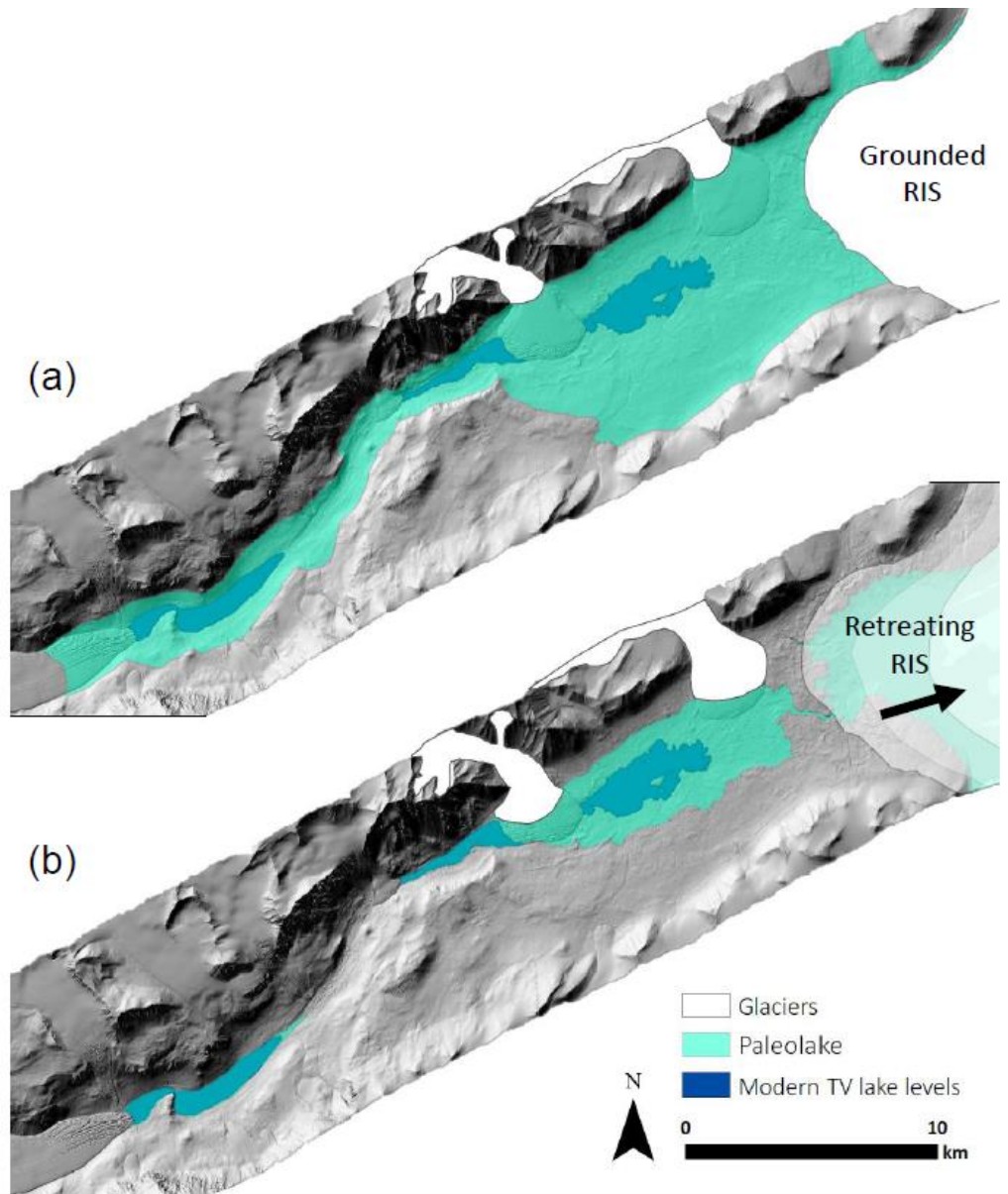

Fig. 3. (a) Glacial Lake Washburn extending up Taylor Valley (300 masl). Extent of grounded Ross Ice Sheet is estimated from maps of Ross Sea Drift moraine locations (Hall and Denton, 2000); (b) Holocene extent of paleolake, limited by 81 masl sill between Lake Fryxell and the McMurdo Sound. Modern day lake levels shown in dark blue. Extent of alpine glaciers (Canada and Commonwealth glaciers in white) are estimated. DEM (1 m resolution) is from 2014-15 LiDAR survey, accessed via OpenTopography (Fountain et al., 2017).


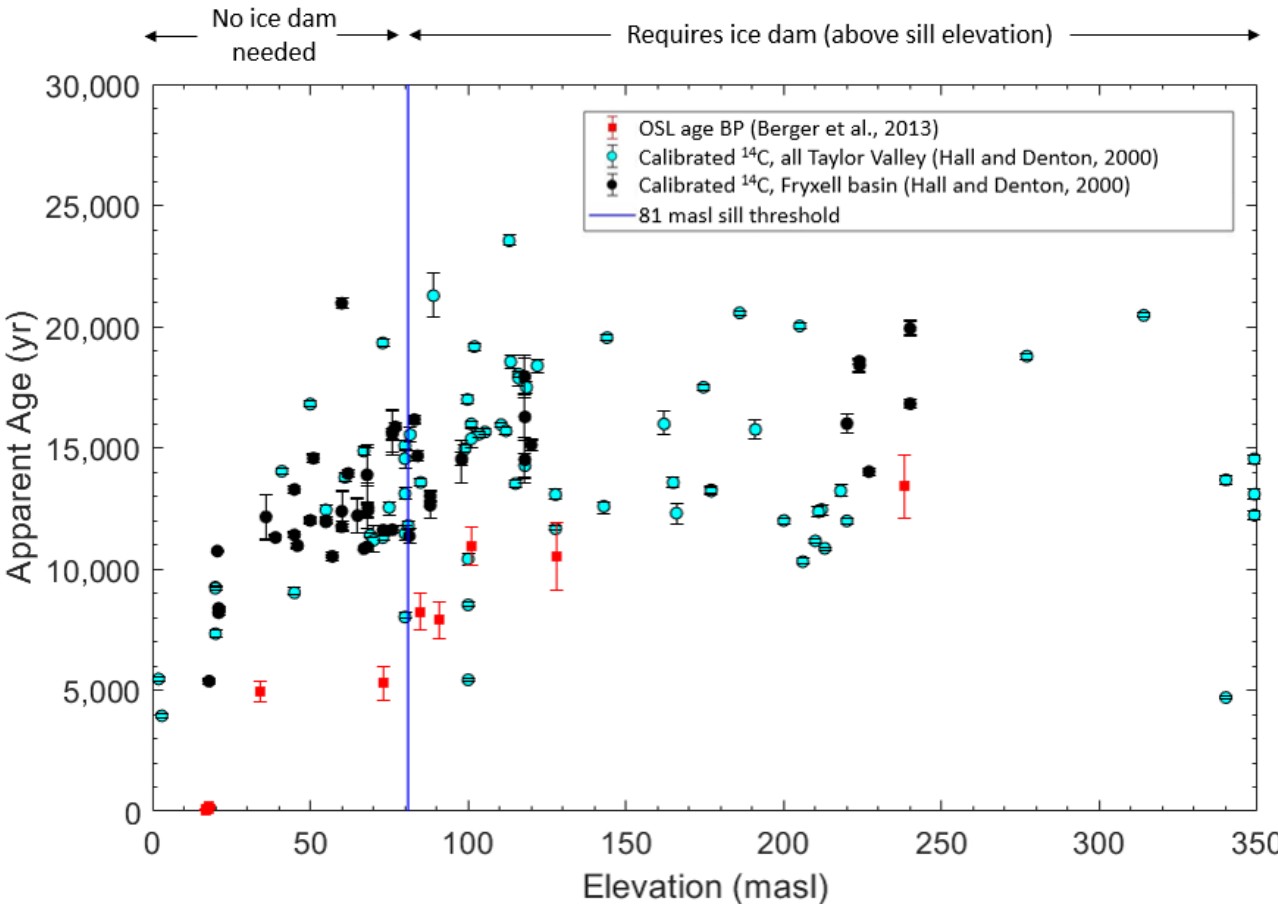

Fig. 4. Age of delta deposits from [14]C dating of preserved algal mats (Hall and Denton, 2000) shown as circles with associated error bars (age). Samples from all of Taylor Valley and just Fryxell basin shown as cyan and black circles, respectively. Age estimates using optically stimulated luminescence (OSL) dating shown as red squares with associated error bars (age) (Berger et al., 2013). The vertical blue line represents approximate elevation of sill at Coral Ridge (81 masl), defining the elevation at which any lake level above would require an ice dam. The calibrated [14]C ages from Hall and Denton (2000) were corrected using the CALIB program (Stuiver et al., 2018).



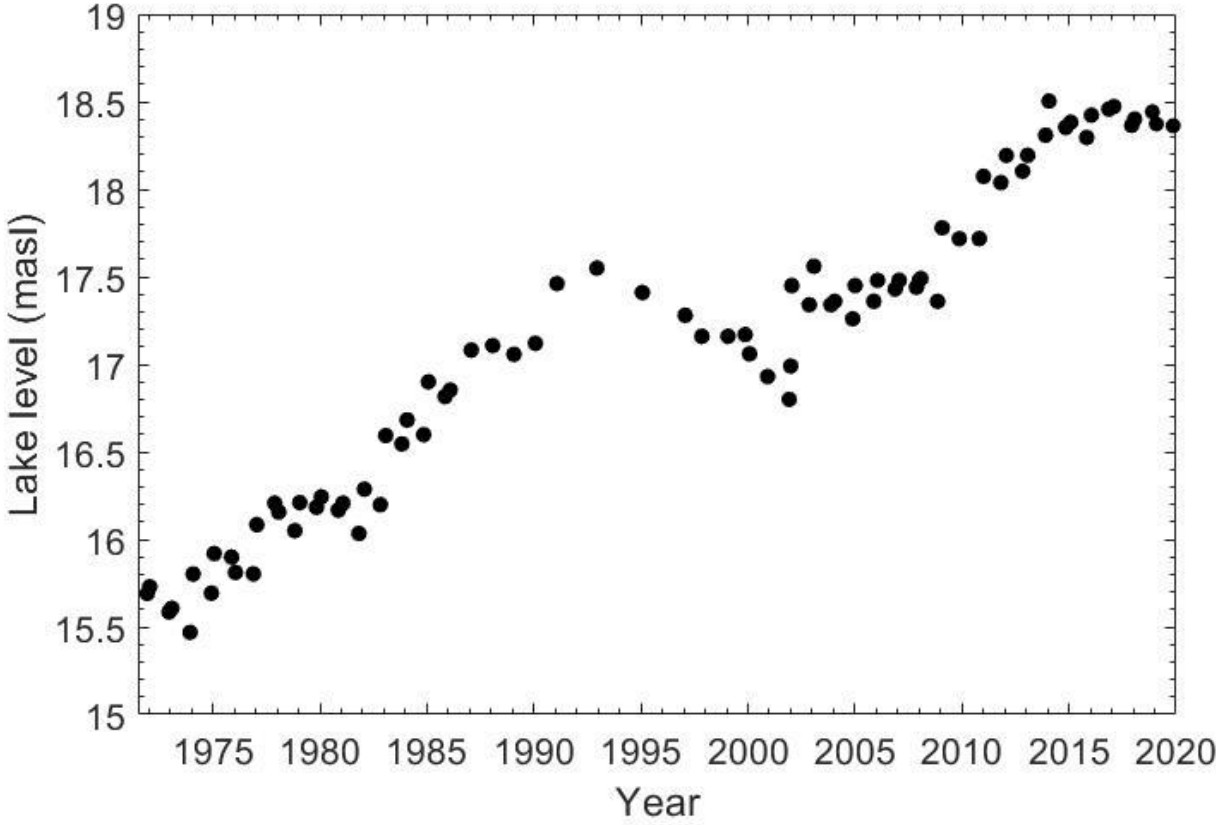

Fig. 5. Lake level elevation in meters above sea level (masl) of Lake Fryxell from 1972 to 2020. Lake levels are determined by manual survey. Data can be accessed at https://mcm.lternet.edu (Doran and Gooseff, 2020).



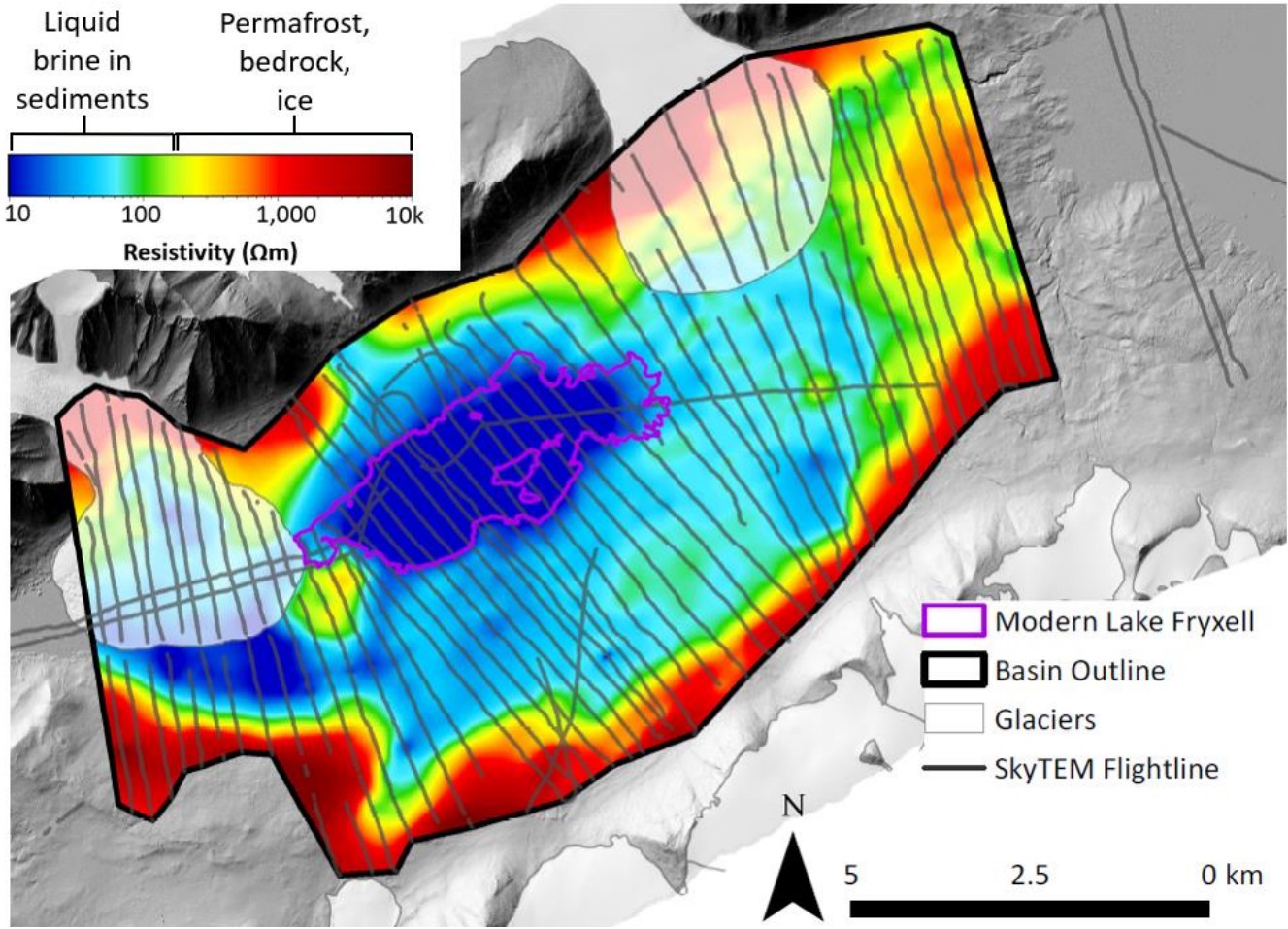

Fig. 6. Mean resistivity map of constant elevation (-100 masl, 5 m thick slice) generated from SkyTEM data. Flight lines are
shown as a transparent grey, which indicate the flight path of the helicopter as it flew over Fryxell basin in 2011. Each flight
line is made up of individual nodes spaced ~25 meters apart. The mean resistivity map was generated in Aarhus Workbench,
and interpolates between nodes to create a 3D resistivity inversion model. This figure represents just one 5 m thick slice of the
3D inversion taken at 100 m below sea level (in the subsurface). The outline of modern lake Fryxell is shown in pink, and
glaciers as transparent polygons. The thick black polygon (Basin Outline) represents the extent of reliable data, which was
truncated at the edge of each flight line to reduce extrapolation artifacts. The digital elevation model (DEM) used for the
background (1 m resolution) is from 2014-15 LiDAR survey, accessed via OpenTopography (Fountain et al., 2017).

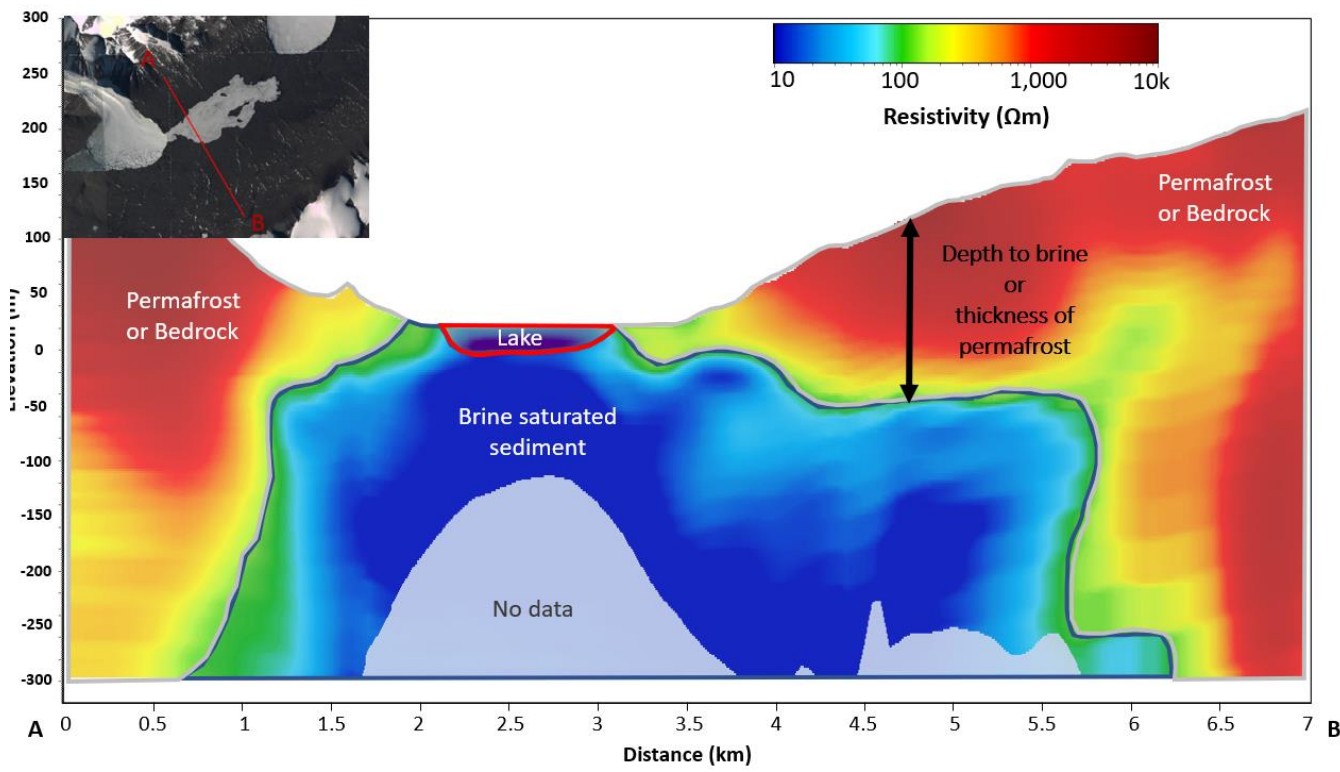


Fig. 7. North-south transect created from SkyTEM resistivity grid in Workbench. Transect location is shown in inset map (A to B) in upper left corner. Mean resistivity maps were created using 1 km search radius (2x line spacing) and 5 m elevation slices. Subsurface boundaries are approximated, and interpretations based on Mikucki et al. (2015). The subsurface boundary

between brine saturated sediments and permafrost is defined using a resistivity threshold of 100 Ωm.


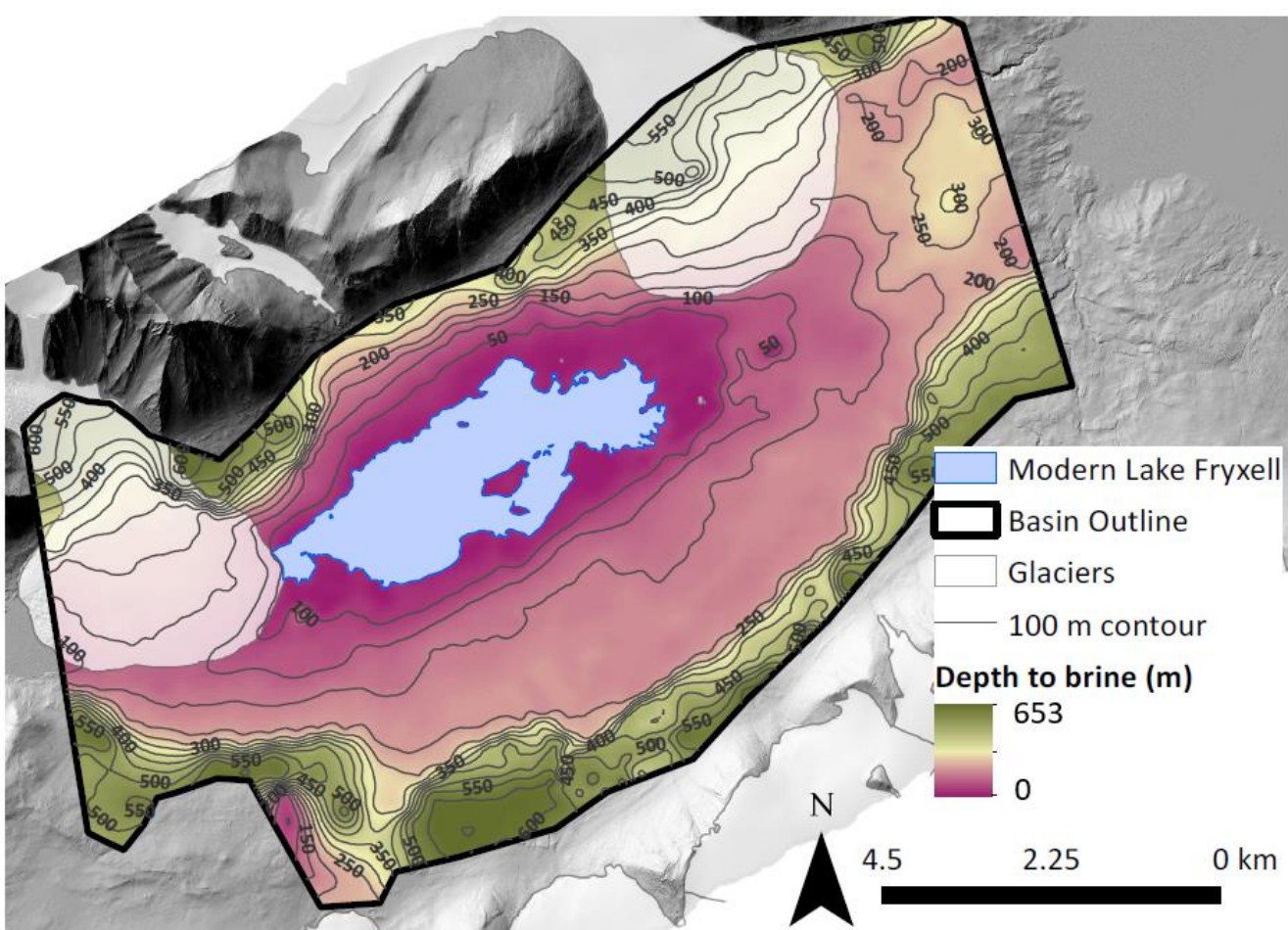

Fig. 8. Map showing depth to 100 Ωm brine/permafrost boundary generated from SkyTEM data. Interpolation was smoothed using a low pass filter in ArcGIS to reduce edge noise. Edges (higher elevations and deeper brine layers) are less reliable because of inversion interpolation artifacts and limitations of depth of investigation. The thick black polygon (Basin Outline) represents the extent of reliable data, which was truncated at the edge of each flight line to reduce extrapolation artifacts. DEM (1 m resolution) is from 2014-15 LiDAR survey, accessed via OpenTopography (Fountain et al., 2017).

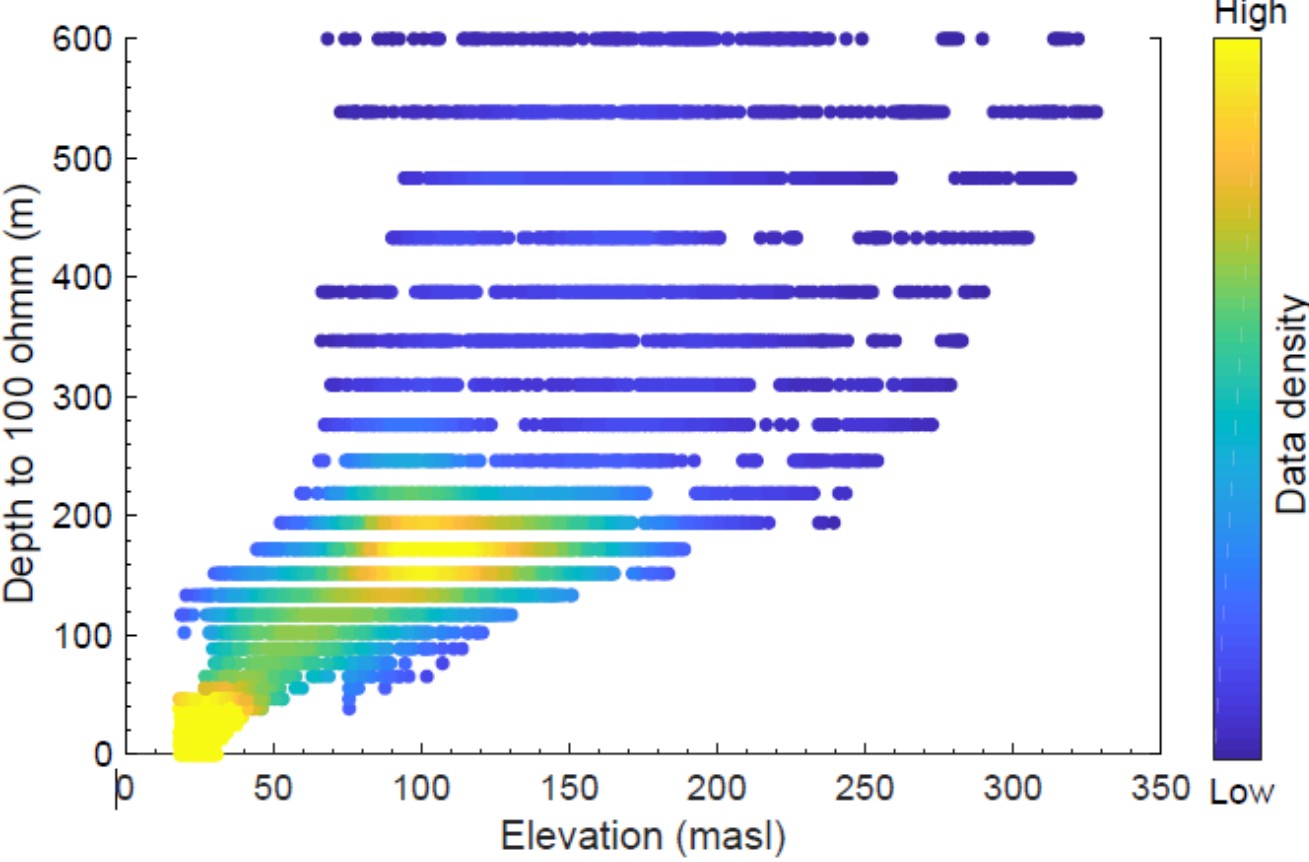

Fig. 9. Elevation in meters above sea level (masl) versus depth to 100 $\Omega$m brine/permafrost boundary in Fryxell basin. Each point represents an individual SkyTEM node (shown as the flight lines in Fig. 6). Data density is displayed as a color ramp, where yellow points indicate higher data density and blue points indicate lower data density.




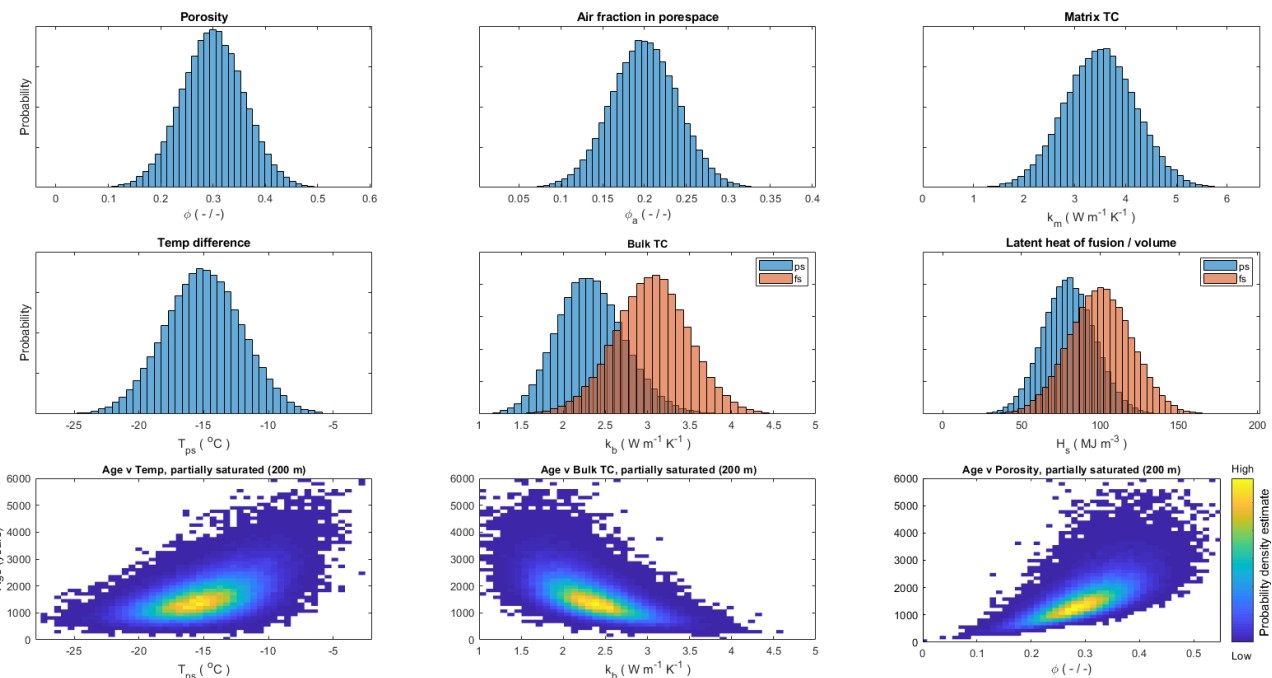

Fig. 10. Monte Carlo sensitivity analysis of permafrost age inputs. $k_b$ and $H_s$ include partially saturated (ps) and fully saturated (fs) substrate conditions. The bottom three panels show how variations in temperature, bulk TC, and porosity affect the age of permafrost. Data density is displayed as a color ramp, where yellow points indicate higher data density and blue points indicate lower data density.

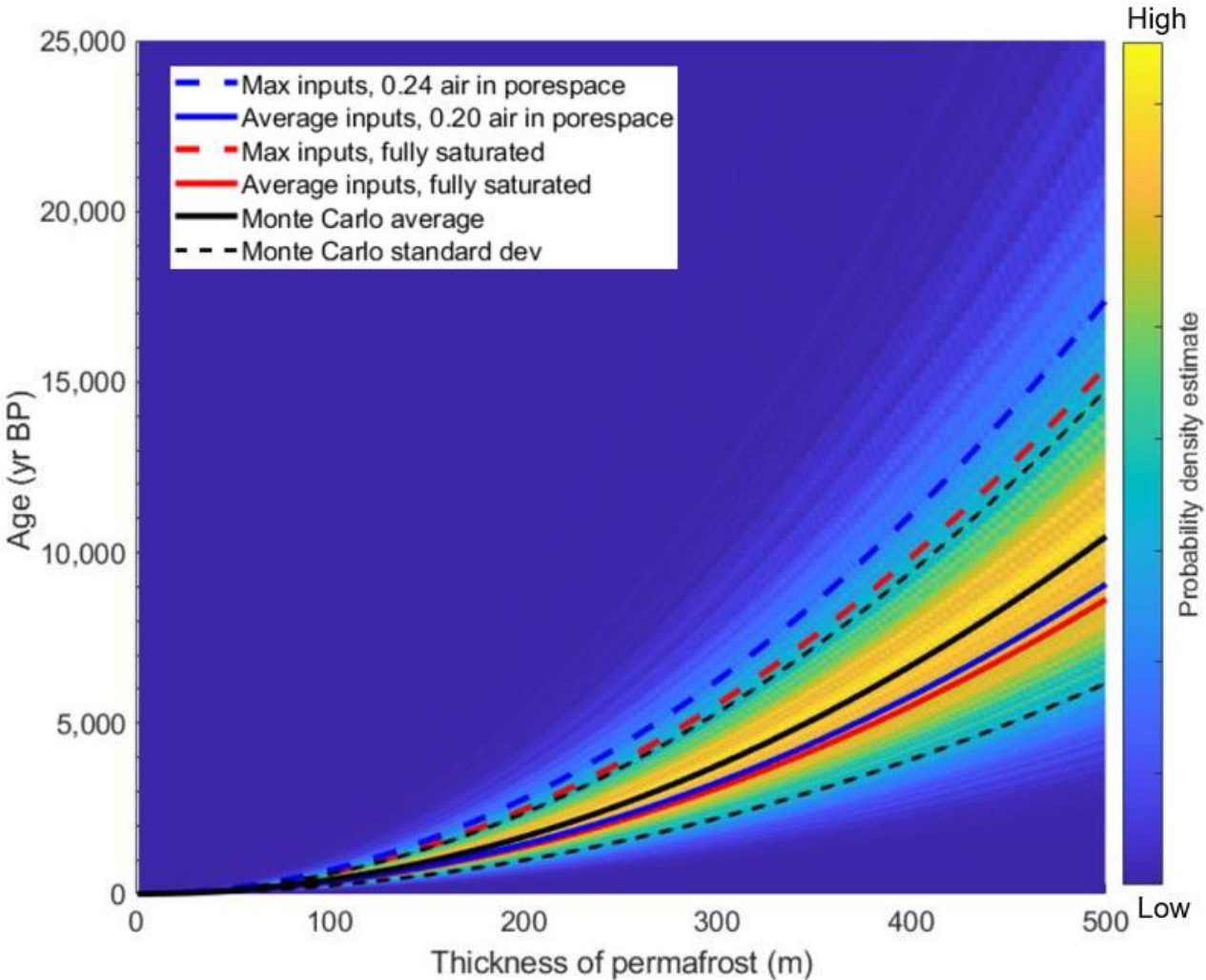

Fig. 11. Monte Carlo age versus elevation probability for 10,000 iterations of Eq. (1) with average inputs and assigned standard deviation. Monte Carlo average result shown as solid black line and one standard deviation as dashed black line. Permafrost ages for average (solid) and maximum inputs (dashed) into Eq. (1), for partially (blue) and completely saturated (red) conditions. Maximum inputs with air fraction in pore space of 0.24 yields maximum possible ages (dashed blue). Data density is displayed as a color ramp, where yellow points indicate higher data density and blue points indicate lower data density.

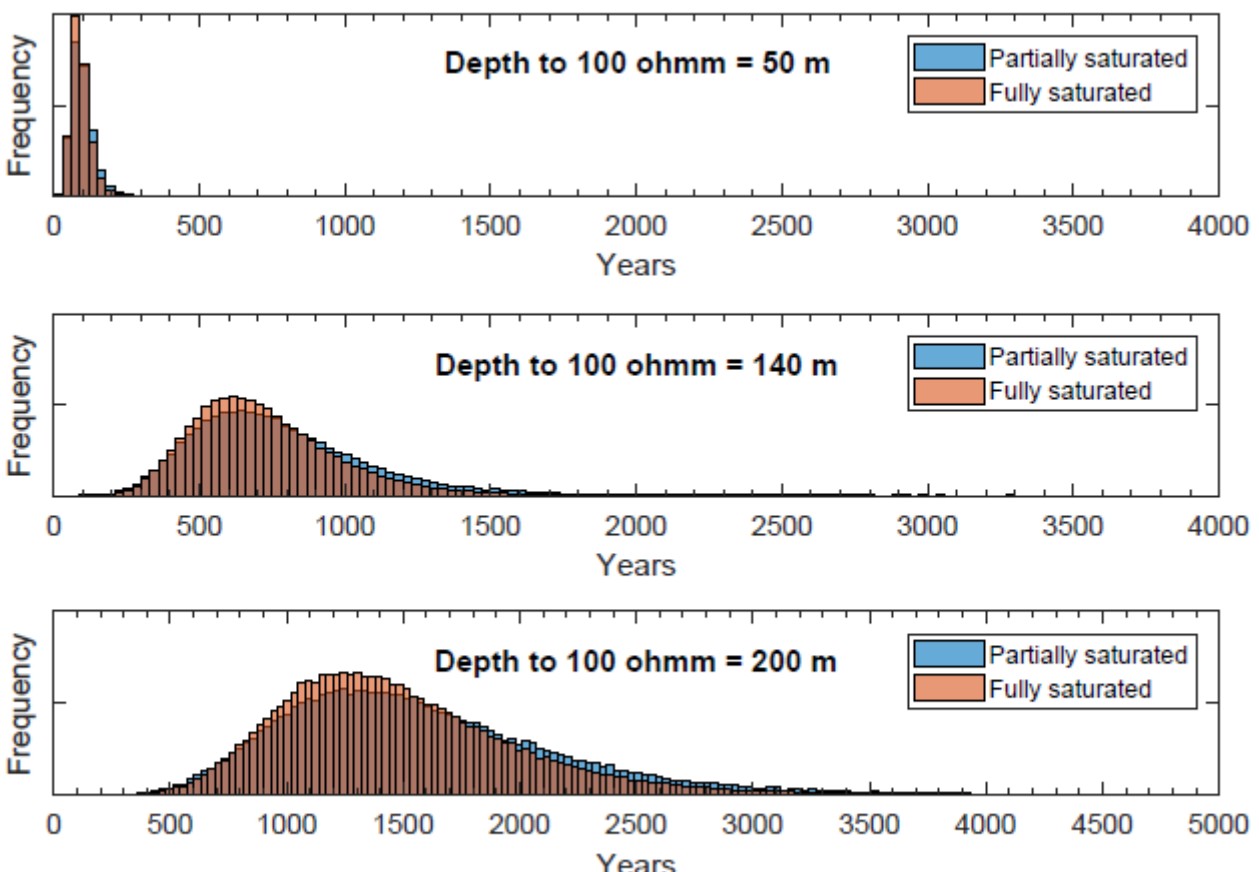

Fig. 12. Age estimates from Monte Carlo analysis as discrete depth to 100 Ωm brine/permafrost boundary. Each panel shows Monte Carlo analysis results for partially saturated (blue) and fully saturated (orange) conditions at three discrete depths to brine. The shallowest depth to brine (top panel, 50 m) shows how age of permafrost is relatively well constrained. Thicker depth to brines (middle panel, 140 m or bottom panel, 200 m) show how increasing depth to brine also results in less constrained ages of permafrost. A partially saturated substrate (blue bars) results in slightly older permafrost ages compared to a fully

saturated (orange bar) substrate.

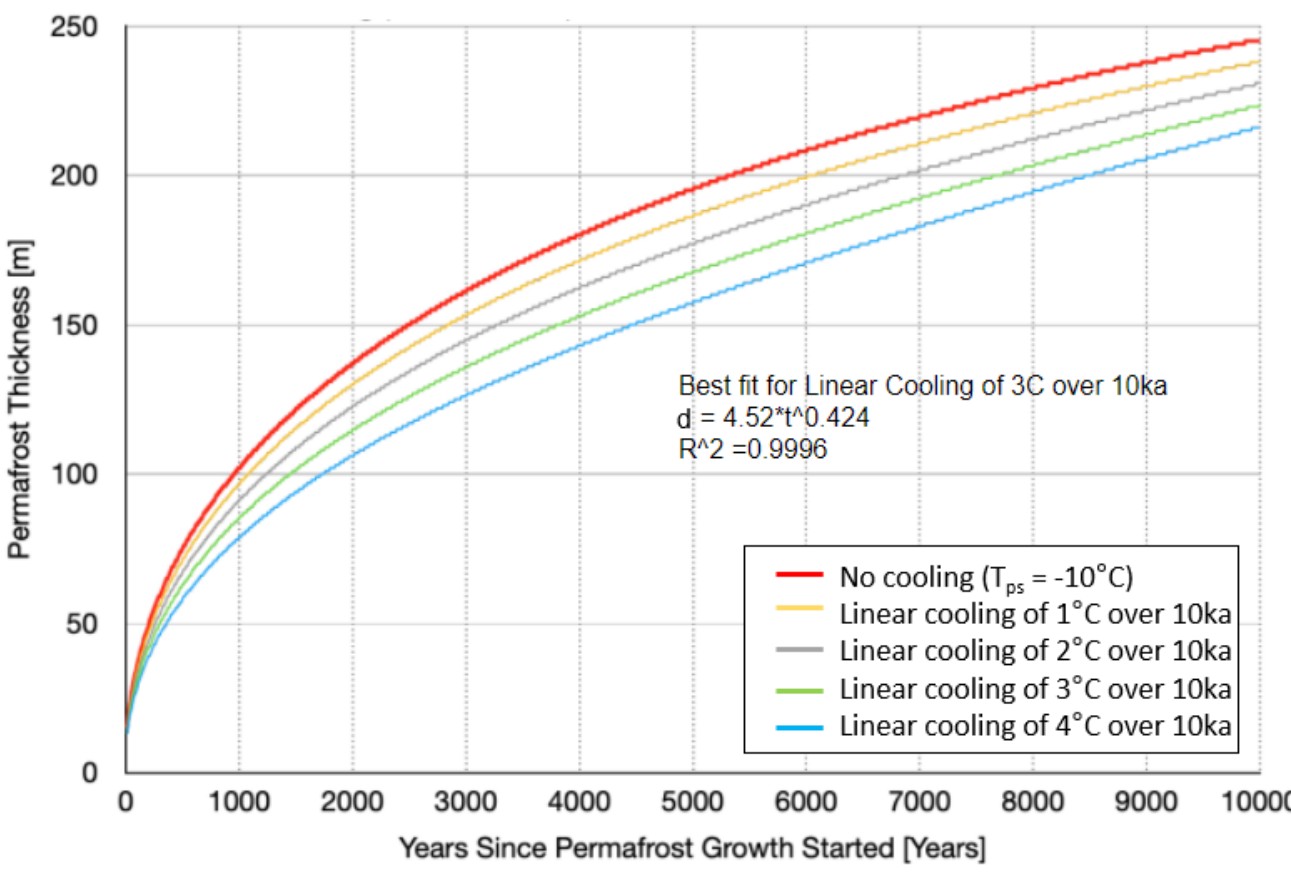


Fig. 13. Results from 1D numerical (finite-difference) model solving the classical Stefan problem of vertical heat diffusion coupled with latent heat release during freezing. The plot shows how age of permafrost increases with increasing permafrost thickness for various linear cooling trends to simulate lowering temperatures in the Holocene. The equation shown above

shows the best fit line for an applied linear cooling rate of 3 °C over 10,000 yrs (green line), Eq. (4).


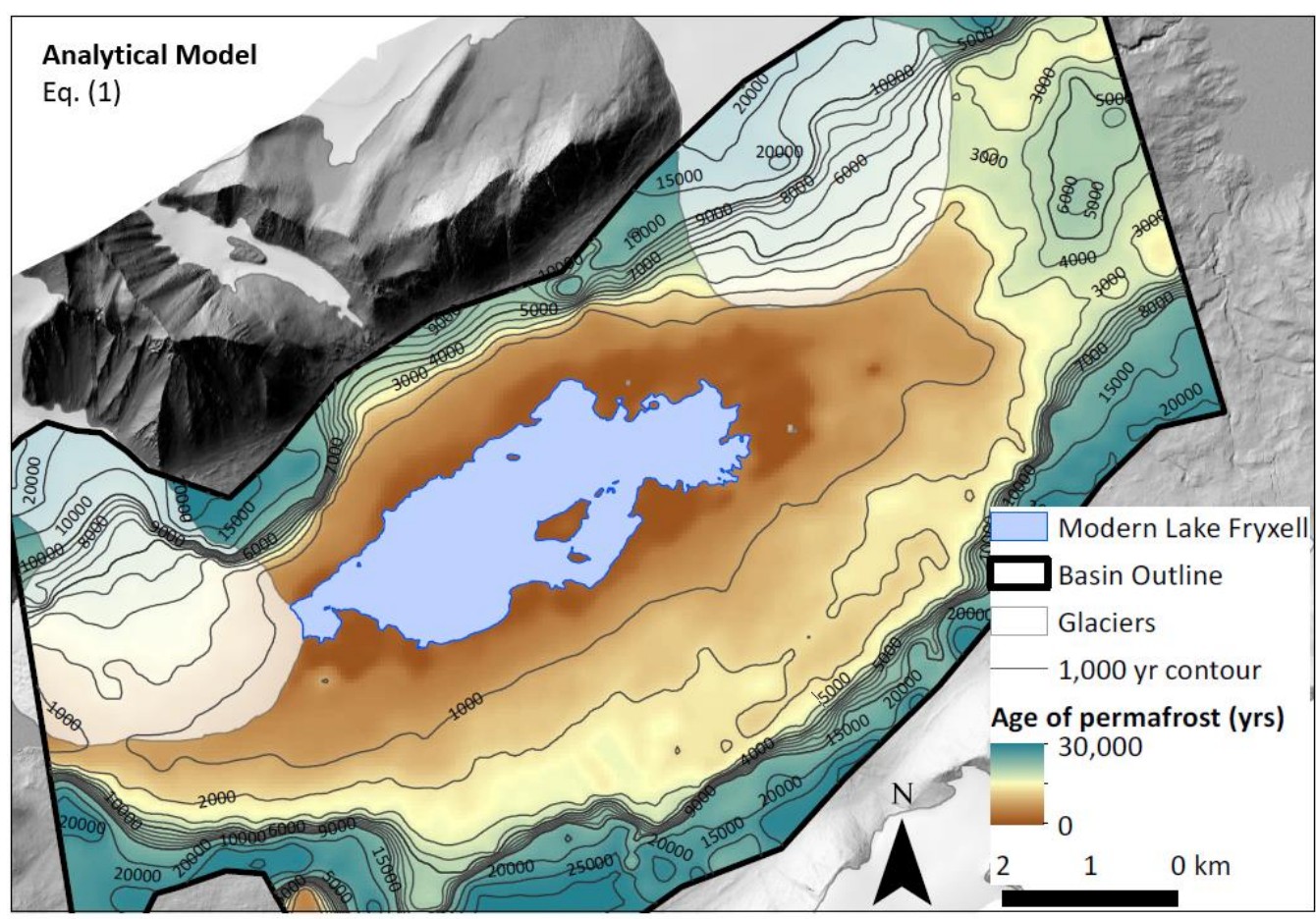

Fig. 14. Analytical model results of maximum permafrost age distribution in Fryxell basin calculated using Eq. (1) (Osterkamp and Burn, 2003). Age is calculated using depth to permafrost-brine boundary (defined as 100 $\Omega$m). Contours represent age of permafrost in years before present, implying higher lake levels (up to 81 masl sill level) during the late Holocene ~1.5 ka. Note that contours switch to 5,000 yr intervals after 10,000 yrs because of such a large permafrost age gradient at high elevations, the 1,000 yr contours were illegible. DEM (1 m resolution) is from 2014-15 LiDAR survey, accessed via OpenTopography (Fountain et al., 2017).

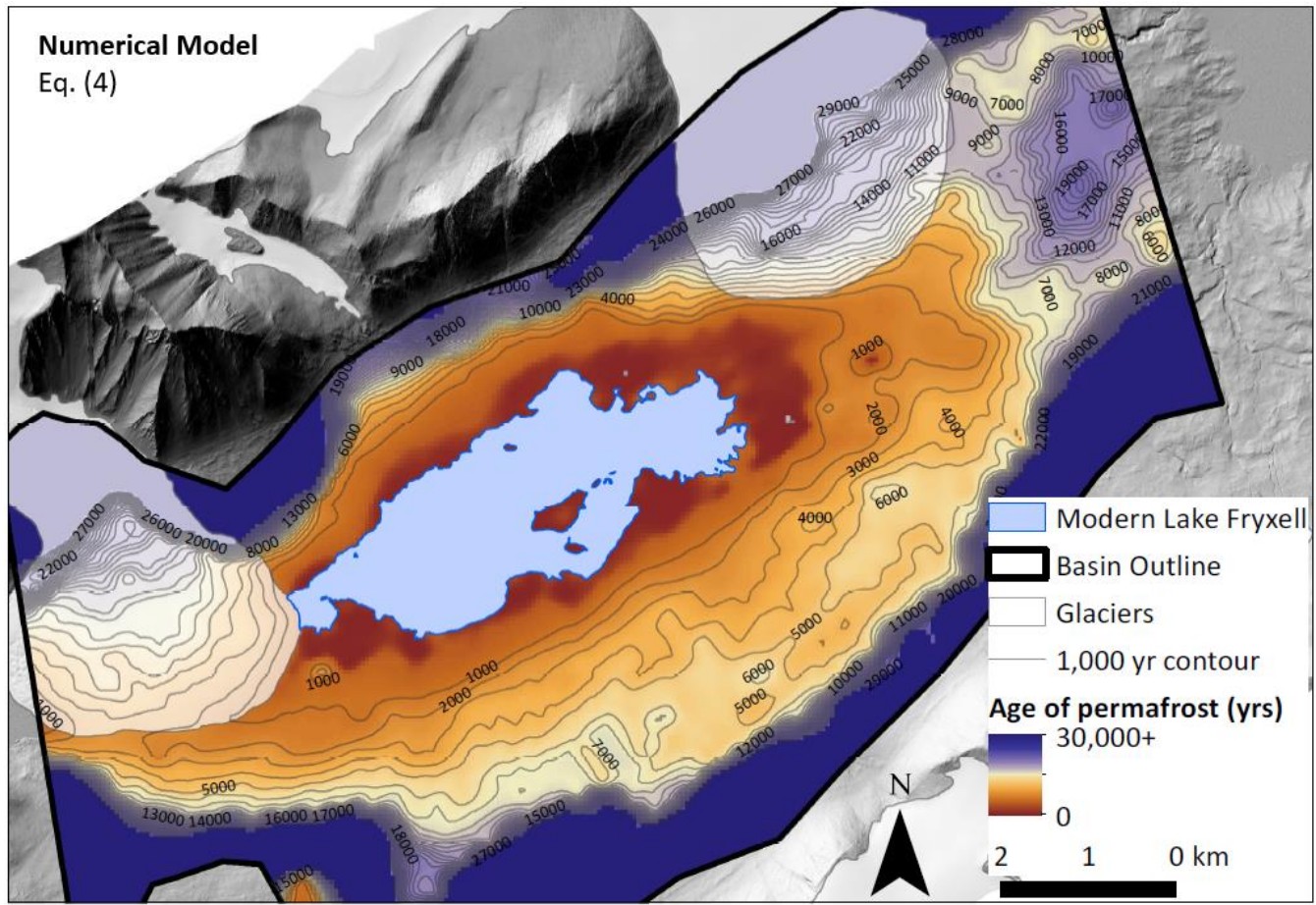

Fig. 15. Numerical model results of permafrost age distribution in Fryxell basin calculated using Eq. (4). Age is calculated using depth to permafrost-brine boundary (defined as 100 Ωm). Contours represent age of permafrost in years before present, implying higher lake levels (up to 81 masl sill level) during the late Holocene ~4 ka, which is older than the results from the analytical model shown in Fig. 14. Note that contours end at 30,000 yr BP because of such a large permafrost age gradient at high elevations contours were illegible. The color ramp also stops at 30,000 yr BP since the model is not accurate for thick (hence older) permafrost. DEM (1 m resolution) is from 2014-15 LiDAR survey, accessed via OpenTopography (Fountain et al., 2017).

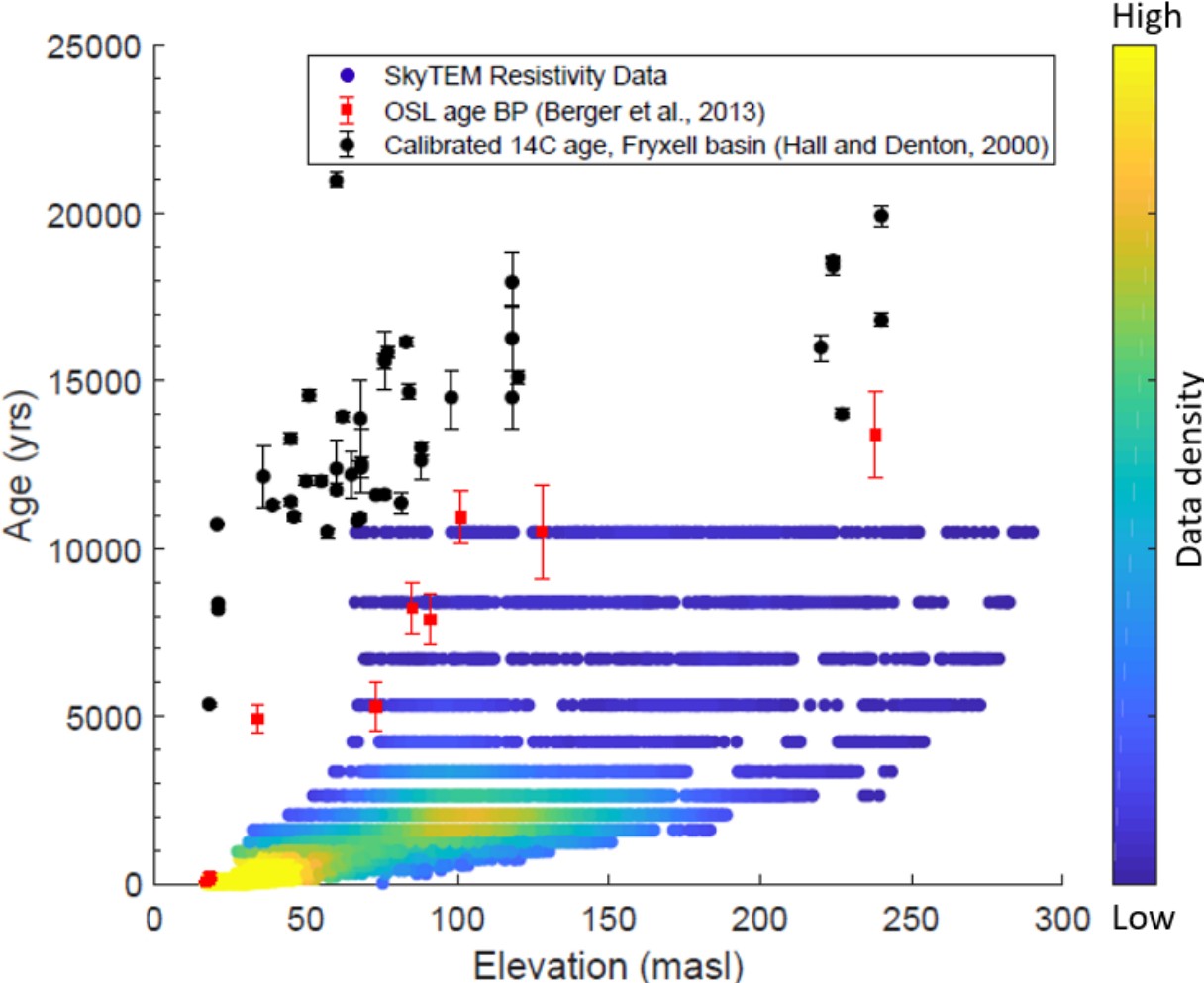

Fig. 16. Permafrost ages calculated using the analytical model (Eq. 1) applied to SkyTEM resistivity data. Depth to brine (100 Ωm brine and inputs from Table 1 were used in Eq. (1) to calculate maximum permafrost age. Permafrost ages are colored by data density, where yellows and greens represent higher data density and blues represent lower data density. Ages are better constrained for younger, shallower permafrost, so ages were cut off at 400 m depth to brine (corresponding to permafrost ages of ~11,000 yr BP). [14]C (Hall and Denton, 2000) and OSL (Berger et al., 2013) ages collected from deltas in Lake Fryxell basin are plotted for comparison.

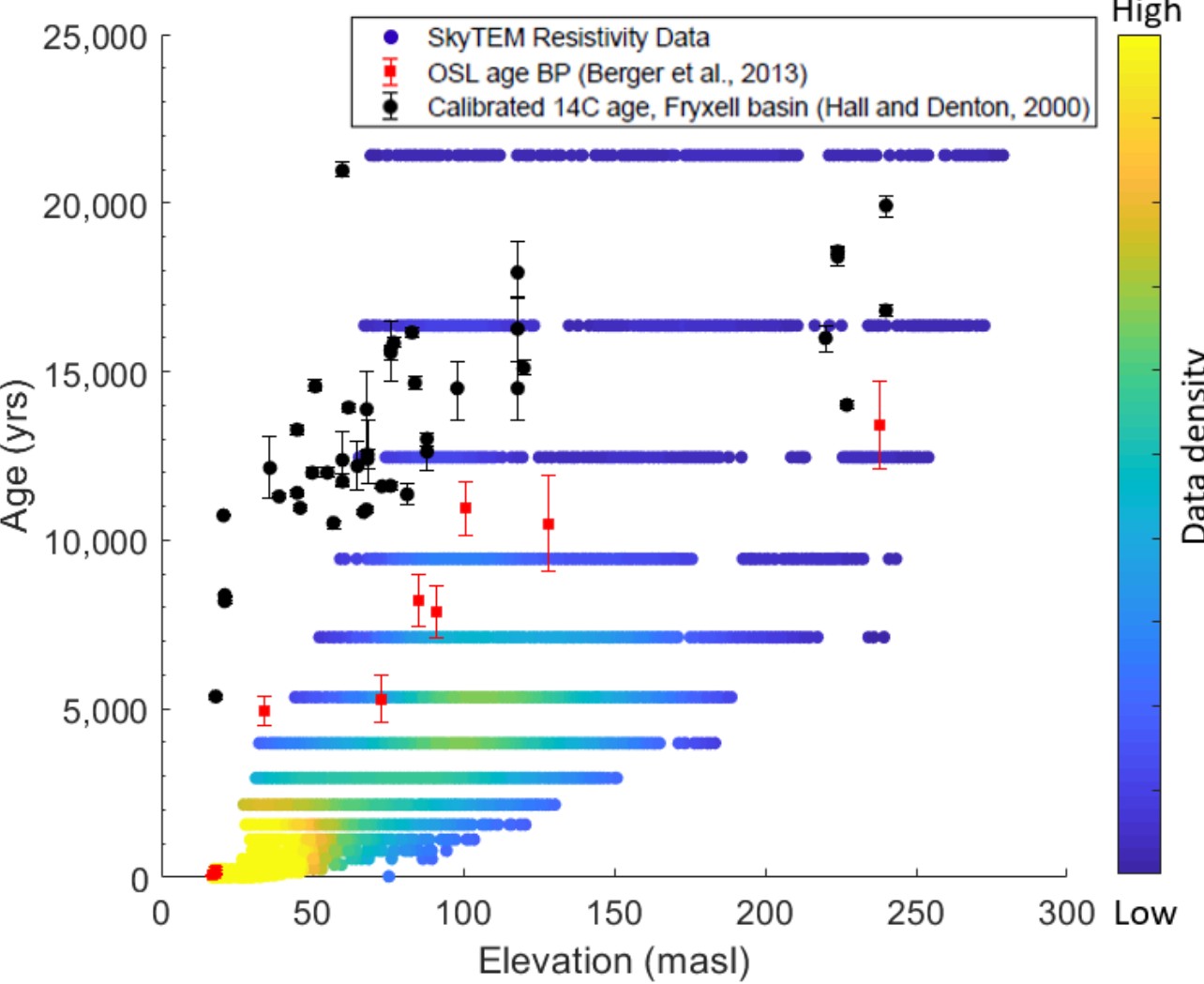


Fig. 17. Permafrost ages calculated using the numerical model (Eq. 4) applied to SkyTEM resistivity data. Depth to brine (100 Ωm brine was used to calculate permafrost age, assuming a 3 °C / 10,000 yr cooling trend over the late Holocene. Permafrost

ages are colored by data density, where yellows and greens represent higher data density and blues represent lower data density. Ages are better constrained for younger, shallower permafrost, so ages were cut off at 400 m depth to brine (corresponding to permafrost ages of ~21,000 yr BP). [14]C (Hall and Denton, 2000) and OSL (Berger et al., 2013) ages collected from deltas in Lake Fryxell basin are plotted for comparison.


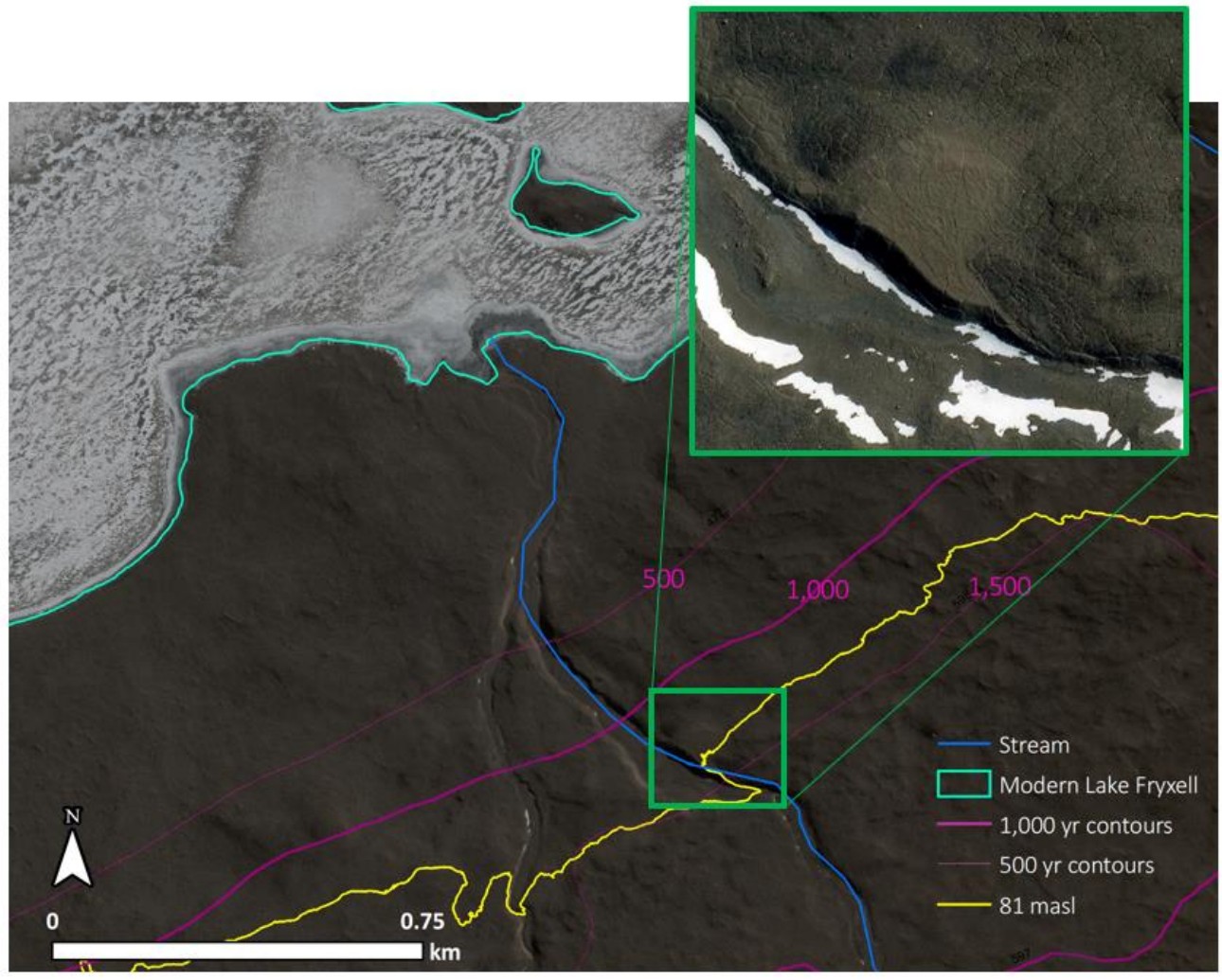

Fig. 18. Maximum permafrost age calculated using the analytical model (Eq. 1) are shown as pink contours and indicate roughly 1.5 ka permafrost at 81 masl (yellow elevation contour). An extremely well-preserved delta on Crescent Stream is located at roughly 81 masl shown in inset map (green box). Image © 2017 DigitalGlobe, Inc. and data provide Polar Geospatial Center.

| Variable | Mean value | Standard deviation | Max/min value | Description |
|---|---|---|---|---|
| $\phi$ | 0.30 | 20 % | 0.36 | Porosity |
| $\phi_a$ | 0 / 0.20 | 20 % | 0 / 0.24 | Fraction of air in pore space |
| $T_{ps}$ | -15 | 20 % | -12 | Temperature difference (K) |
| $k_f$ | 2.25 | 2 % | 2.205 | TC of fluid (-5 °C) (W/m K) |
| $k_a$ | 0.0236 | 2 % | 0.0231 | TC of air (-10 °C) (W/m K) |
| $k_m$ | 3.50 | 20 % | 2.80 | TC of rock matrix (-5 °C) (W/m K) |
| $k_{b, fs}$ | 3.07 | - | 2.57 | Bulk TC (W/m K), fully saturated |
| $k_{b, ps}$ | 2.33 | - | 1.73 | Bulk TC (W/m K), partially saturated |
| $H_{s, fs}$ | 100.2 | - | 120.2 | Vol. heat of fusion (MJ/m$^3$), fully saturated |
| $H_{s, ps}$ | 80.2 | - | 91.4 | Vol. heat of fusion (MJ/m$^3$), partially saturated |


Table 1. Input values for Eq. (1) through Eq. (4) and Monte Carlo analysis to calculate permafrost ages. Max/min values column show values within one standard deviation resulting in oldest permafrost ages.

