# Peer review of "Thermal legacy of a large paleolake in Taylor Valley, East Antarctica as evidenced by an airborne electromagnetic survey"

_The Cryosphere, 2020_

## Referee Comment (RC1) · Anonymous Referee #1 · 19 Nov 2020

This manuscript presents a novel approach based on airborne transient EM resistivity surveys and permafrost refreeze modeling to reconstruct the recent (<8 ka) paleo-hydrological history of the Lake Fryxell basin in the McMurdo Dry Valleys. The resistivity data collected within the Lake Fryxell basin show a clear signal of subsurface brine and permafrost distribution, which is analyzed to provide a maximum age for the last lacustrine draw down event of 1-1.5 ka. Permafrost depth and refreeze modeling suggest that following the ice sheet retreat at 8 ka, lake levels likely fluctuated to up to 81 m above sea level until 1.5 ka. These results provide new insight and place new constraints on recent groundwater and lake level variability that were not detected by other techniques.

I have only one major comment regarding the assumptions made in permafrost modeling. As acknowledged in the discussion section, "this model assumes a constant rate of lake level drop and constant Tps for simplification." I strongly recommend including a dedicated paragraph to discuss the ramifications of these assumptions and how results may be affected. For example, how much would the maximum permafrost age change if Tps was allowed to vary by an extra 2, 3, 5 K? What would be the effect on permafrost growth at depth if the ice dam partially collapsed in one or more episodes instead of allowing for a more gradual draw down?

——————- Minor comments below, indexed by line and figure number ——————-

General comment on acronyms: the manuscript contains a lot of acronyms, which affect the readability for readers who are less familiar with the region and/or techniques in this study. Some acronyms are only used a handful of times, such as RIS, GLW, AEM, DOI, and DVDP, and thus I suggest spelling out the entire words instead. "LGM" appears to be used only four times but is a well-known acronym and I feel it can be left as it is. Also, there are two acronyms that are not spelled out: 49: TV - Taylor Valley? 118: DVDP - Dry Valley Drilling Project?

102: What type and parameters of kriging interpolation was used? Also, why was kriging preferred over other interpolation techniques? Kriging is a predictive algorithm and may diverge or create artifact under certain conditions. I believe the authors need to provide some information regarding the configuration of the kriging interpolation and motivate the choice over other algorithms.

113: I suggest adding some information on the DEM employed in this study.

134: Here the authors use the -20 °C average air temperature of Lake Fryxell from Obryk et al. (2020) to calculate the age of permafrost. However, this temperature was calculated over a timespan of 30 years, and thus may not be representative of the air temperature since the permafrost refreeze initiation. I understand that the Monte Carlo analysis takes the uncertainty of each parameter into account, but I think there should

be a discussion on the reliability of a recent temperature measurement in the context of a much longer time scale.

145: Is a geometric mean appropriate to calculate the bulk thermal conductivity of sediment, fluid, and air mixtures in this scenario? I recommend motivating the usage of a geometric mean over other mixing formulas. For example, Fuchs et al. (2013; https://doi.org/10.1016/j.geothermics.2013.02.002) explore a few different mixing formulas and find that some are better than others for specific sediment mixtures.

152: I recommend writing either "variance" or "standard deviation." As it is written in the manuscript, it seems like the two are the same thing.

General comment on the results section and related figures: I suggest moving or at least copy some of the text in the result section over to the caption of relevant figures. Currently, the captions are on the minimalistic side, and I believe that adding further explanations would greatly improve the readability of the paper when readers glance through it quickly.

247: Inherent -> inherit ?

Fig. 6 and 7: The usage of this rainbow color scale is problematic for a couple of reasons. (1) It is visually non-linear, with sudden jumps in hue that may result in apparent variability of the dataset that does not actually exist. For example, there is a large jump in light blue-green-yellow that conveniently coincides with the proposed boundary between brine and permafrost resistivities; although this helps locating such putative boundary, I find it potentially misleading. (2) It is very hard to read by colorblind people. To the most kind of color blindness cases, this color scale looks symmetrically identical below and above 200 ohm*m, thus making it very difficult to distinguish which areas are low and high resistivity. Fig. 8 and 13 also employ a non-linear color bar with a large jump mid-range.

---

## Referee Comment (RC2) · Anonymous Referee #2 · 5 Jan 2021

My first impression upon reading this paper was, wow, what a novel way to approach this problem! People have been trying to track Dry Valleys lake levels since Scott in the Heroic Age. None of the existing datasets is complete. Landforms (mostly deltas) dated by radiocarbon tell you when the lake was at a certain level, but geomorphic records are fragmentary and incomplete. Lake bottom sediments from beneath the present lake can give more continuous records, but their interpretation in terms of specific water level is complicated. Here, the authors give a third way of reconstructing former lake levels, which I'm sure has its own associated problems, but which presents a third perspective that may help us circle ever closer to the actual lake history.

[Figure]

That said, I cannot judge the actual methods used in the remote imaging or in interpretation of the resistivity data. It is outside of my field of expertise. I am assuming here that it is correct, and more reliance should be placed on other reviewers on this aspect.

Although my overall impression of the paper is quite favorable, I do think there are some areas that could use reframing and a critical assumption or two that should be revisited.

Major Points 1) The analysis seems to operate on the assumption that there was a continuous drop in lake level from the ~80 m sill level to the present lake over the second half of the Holocene (and indeed, perhaps from the highstand of GLW). However, this is unlikely to have been the case. This would have been a closed-basin lake and highly sensitive to changes in inputs (almost entirely glacial meltwater) and outputs (evaporation/sublimation). This hydrology places it among a category of lakes that typically shows outsized reactions to small changes in water balance. Pre-Holocene (and even Holocene) lakes in the Dry Valleys region are suggested to have undergone large-scale fluctuations during the last glaciation and termination of the ice age (i.e., Hall et al., 2010, PNAS - a reference which probably should be included). Lake Vanda showed large fluctuations at ~3 and ~6 ka. So, it is likely that Lake Fryxell did as well - and possibly other changes not yet documented. To what degree does this lake-level volatility affect your modelling results and error bars? How long must permafrost be covered up to completely melt under a lake (I assume this results from permafrost thickness and length of lake cover)? In short, it seems as if this assumption may be quite critical to the outcome of the model.

2) The paper attempts to compare model results with 14C and OSL data. However, this is mixing of apples and oranges (and may stem from the assumption in point 1 above). Both the 14C and OSL dates refer to the position of the lake at a specific time. They do not preclude the lake from reaching that same level (or higher) at another, later time. It is highly likely that most if not all of the dated deltas were under water at multiple times, not necessarily reflected by the dates (this is to some degree acknowledged in the

discussion of OSL, but the same applies to radiocarbon). Some deltas at critical levels (i.e., the sill level) may have been occupied and active at several times. Thus, in my opinion, you cannot make a direct comparison of the three different dating methods, simply on elevation. These lakes fluctuated many times and there are going to be deposits from earlier times mixed in at the same elevations occupied by later lakes. Recognition/resolution of this assumption may help resolve some other issues raised below.

3) I am having some difficulty with Fig. 14. If I am understanding it correctly, parts of the modelled curve become problematic and require some explanation. I have already mentioned the 14C and OSL data - I don't think you can compare them directly for reasons mentioned above and recommend taking them off this diagram. My concern here is about the model results themselves.

My understanding is that this figure shows the mean of the model (solid line), a dark shaded zone (1 sigma), and a light shaded zone (raw data). The underlying assumption here is that the permafrost age is directly linked to the lake presence (let's revisit this in a minute). Thus, the older permafrost ages at higher elevations are attributable to the lake dropping from there earlier. Those at >80 m must be from a time when there was still an ice dam at the valley mouth, because they are higher than the sill. However, model ages for >80 m seem far too young. Measurements by multiple methods place deglaciation at the mouth of Taylor Valley by ∼8 ka if not a bit earlier. On Fig. 14, 8 ka intersects the model results (solid line) at ∼160 m elevation, something that is not possible, because there would have been no ice to dam the lake. Even the 1 sigma error bars are well above the 80 m sill. The lake cannot exist above 80 m after 8 ka, so how is this result explained? Only the low end of the raw data fit in the plausible zone (>80 m = >8ka). Is there something about the model or in the picking of 100 $\Omega$m that is producing results skewed toward young ages? How does this affect the reliability of the conclusions in this paper? In other words, if we know the model is producing erroneous ages above 80 m, why should we have confidence in results below 80 m

[Figure]

elevation? Greater discussion would be useful here.

4) While it seems plausible, is changing lake level the only variable that can affect permafrost age? How old is that brine?

Minor Points (by line) Line 49 - use the calibrated value, not the raw 14C age.

Line 54 - The Taylor Dome Holocene chronology has been revised (Monnin et al., 2004), which actually is fairly substantial in the mid-Holocene. Use of any specific times should be checked as, if my memory is right, parts of the chronology have shifted as much as ∼1500 years.

Line 61 - "atmospheric correction" is an odd way to put it. I would simply call them calibrated ages. Hall et al., 2010 in PNAS is a key paper here with an updated dataset (see the SI with that paper).

Line 67 - I'm not sure the lack of samples led to an assumption that lakes remained at or below modern levels over the last 5000 years but only that there were no data and that Lake Fryxell had to be below the sill.

Line 106 - says <1000 Ωm but line 115 and elsewhere uses 100 Ωm as the cutoff. Is this a typo or something else? If not a typo, why was 100 chosen?

Line 196 - see comment above about these deltas being covered and recovered by water and thus direct comparison is impossible.

Line 205 - this distance would be made more useful for lake level if an elevation were associated with it.

Line 213 - this assumption is critical and might be key.

Line 225 - this paragraph goes with the assumption of gradually lowering lake, which was not the case. See Hall et al., 2010. It also makes the assumption of a close link between air temperature at Taylor Dome and lake level, which may be problematic. Also, see comment above about revised Taylor Dome Holocene timescale.

Line 238 - I may have missed it, but I didn't see anywhere in Hall and Denton (2000) where they stated that lake levels had remained at or below present since 5 ka. They didn't have data to address this.

Line 243 - the sentence about OSL dates from depth and not applying to most recent episode of lake level change applies equally to radiocarbon.

Line 249 - application of a large reservoir correction to shallow water lake sediments seems ad hoc, given prior references. Most of these deltas formed at the same stream mouths as today, fed locally by the same glaciers as today, not the RIS. As the data should not be compared directly (for reasons already given), such an attempt at correction is not warranted.

Line 259 Whittaker here and later in paragraph.

Par. starting with 265. While I don't necessarily disagree with the conclusion of the last drop in lake level being post 1.5 ka, I find these stated reasons unconvincing. The link with the Taylor Dome ice core here implies, without specifically stating, that high temperatures should lead to high lake levels. But, this may not be the case. If you want to make this argument, this needs to be stated explicitly (and you'll want to confirm that the chronology didn't shift too much for this to still apply). In addition, there are many, large, well-preserved deltas and while the Crescent Stream one is a nice one, weathering rates are so low that you cannot resolve 1000 yrs vs 10,000 years based on delta appearance.

Be careful with the 6 degrees warmer comments about Taylor Dome. Temperature is only one of several variables that affect stable isotope data. In addition, the 100 yr running average in the Steig paper (probably more applicable to lake-level changes than any single year) suggests only about ~1 degree change and is probably a better representation of actual temperature changes.

I think these reasons don't help your argument any. Why not just say your data suggest

the lake might have last been at the level of the sill at 1.5-1 ka and leave it at that? To my knowledge, there is nothing that says it couldn't have been. The OSL and 14C data are older, but they may be dating earlier events.

Line 287 and elsewhere. GLW refers only to the ice-dammed lake. Once the ice sheet retreated, this became Lake Fryxell.

Please double check references. I didn't check them all, but the one I looked up to make sure it was spelled correctly (Whittaker) wasn't there.

Figures 1) Please add latitude/longitude 2) You may want to switch to the Monnin et al., 2004 timescale. 3) May want to mark the location of this figure on Fig. 1 4) suggest changing "atm corrected" to "calibrated" 6) I found the color on these maps (this and the ones that follow) hard to follow, I'm afraid. Is there a better way to present this? The caption could be more informative for those (like me) who do not deal in resistivity commonly. Perhaps you could add a sentence about what is meant by constant elevation. 13) see comment above about using GLW. This should be "lowering of Lake Fryxell" rather than draining of GLW, which would have happened when the ice sheet retreated. Draining really isn't the right word for something that evaporated. 14) was covered in comments above. Suggest removing other chronologic data as the comparison doesn't really mean anything and is potentially misleading.

Overall Impression Although I have pointed out several potential issues with the paper, I think the approach is really intriguing and the study is important to publish. I think most of these comments can be dealt with fairly quickly. Assessing the impact of the assumption about constant lake-level drop (which is critical), may take more effort. I recommend publication after moderate revision.

---

## Author Comment (AC1) · 4 Mar 2021

The authors first want to thank the reviewer for their comments which will greatly approve the manuscript. We appreciate it!!

Reviewer Comment (RC) 1) This manuscript presents a novel approach based on airborne transient EM resistivity surveys and permafrost refreeze modeling to reconstruct the recent (<8 ka) paleohydrological history of the Lake Fryxell basin in the McMurdo Dry Valleys. The resistivity data collected within the Lake Fryxell basin show a clear signal of subsurface brine and permafrost distribution, which is analyzed to provide a maximum age for the last lacustrine draw down event of 1-1.5 ka. Permafrost depth

and refreeze modeling suggest that following the ice sheet retreat at 8 ka, lake levels likely fluctuated to up to 81 m above sea level until 1.5 ka. These results provide new insight and place new constraints on recent groundwater and lake level variability that were not detected by other techniques. I have only one major comment regarding the assumptions made in permafrost modeling. As acknowledged in the discussion section, "this model assumes a constant rate of lake level drop and constant $T_{ps}$ for simplification." I strongly recommend including a dedicated paragraph to discuss the ramifications of these assumptions and how results may be affected. For example, how much would the maximum permafrost age change if $T_{ps}$ was allowed to vary by an extra 2, 3, 5 K? What would be the effect on permafrost growth at depth if the ice dam partially collapsed in one or more episodes instead of allowing for a more gradual draw down?

Author Comment (AC) 1) We appreciate this comment from the reviewer, and agree that this topic deserves more discussion. We have decided to include a second approach to calculating permafrost ages using a 1D numerical (finite-difference) model solving the classical Stefan problem of vertical heat diffusion coupled with latent heat release during freezing (see attached Figure A). The upper boundary condition is a prescribed temperature that is deltaT lower than the freezing point of the sub-permafrost brines (deltaT = surface temperature - brine freezing temperature). This deltaT can be either held constant during numerical experiments or can be prescribed to vary with time. The bottom boundary condition of the numerical model is a constant heat flux, set to the geothermal flux of 0.080 W/m2 consistent with two borehole-based estimates proximal to the study area (boreholes DVDP-6 and CIROS-1 in table 1, Morin et al., 2010). Other model parameters are based on permafrost properties listed in Table 1 in our original manuscript.

Existing observational constraints indicate that under modern conditions in the study area, the temperature at the bottom of the permafrost is ca. -9C (e.g., Figure 7 in Foley et al., 2015) while ground surface temperature is ca. -19C (Table 1 in

[Figure]

Obryk et al., 2020), yielding deltaT of about 10C. When we assume a constant deltaT of 10C, our numerical experiment still yields fairly young ages for elevations below the 81 masl sill level (<4ka). We then applied a linear cooling rate of 1, 2, 3, and 4C over the last 10,000 yr to model the cooling trend observed in the Holocene Taylor Dome paleotemperature reconstructions (Steig et al., 2000, Monnin et al., 2004). These ice core constraints are best approximated by a linear cooling trend of 3C per 10,000 years. In the revised version of the manuscript we plan to use these results to create an additional figure (similar to Figure 13) which shows the permafrost age distribution using the 1D vertical diffusion model. This will better address the reviewer's concern of assuming a constant Tps throughout time.
* * *
Minor comments below, indexed by line and figure number

RC 2) General comment on acronyms: the manuscript contains a lot of acronyms, which affect the readability for readers who are less familiar with the region and/or techniques in this study. Some acronyms are only used a handful of times, such as RIS, GLW, AEM, DOI, and DVDP, and thus I suggest spelling out the entire words instead. "LGM" appears to be used only four times but is a well-known acronym and I feel it can be left as it is. Also, there are two acronyms that are not spelled out: 49: TV - Taylor Valley? 118: DVDP - Dry Valley Drilling Project?

AC 2) We agree that it would be useful to spell out some acronyms that are only used a few times. We made these changes to the manuscript.
* * *
RC 3) Line 102: What type and parameters of kriging interpolation was used? Also, why was kriging preferred over other interpolation techniques? Kriging is a predictive algorithm and may diverge or create artifact under certain conditions. I believe the authors need to provide some information regarding the configuration of the kriging interpolation and motivate the choice over other algorithms.

AC 3) The software used for data processing, Aarhus Workbench, currently has the option of inverse distance or kriging interpolation, with actual calculations being carried out by gstat (Pebesma, E. J. 2004 https://www.sciencedirect.com/science/article/pii/S0098300404000676#aep-section-id12). Kriging involves the use of a semi-variogram to determine weights during the interpolation, which makes this method well-suited to capturing spatial correlations, both methods result in similar images in the main area of interest. The variogram model we use in the kriging is a simple exponential function with log-transformed resistivity values, a sill value of 0.16 and a range of 1520 m. We will add a segment about why we chose kriging to the manuscript in Section 2.1 (Resistivity Surveys).

We also added a sentence to further highlight some of the artifacts of the interpolation that are not real, especially around the edges of the mean resistivity map (Figure 6). We added model node locations to our location map to provide an easy visualization of data density, which will give insight into which features may be anomalous artifacts and which are model based. When making Figure 6, we had to make a tough decision defining interpolation search radius (here we chose 1,000 m) in order to allow enough overlap between surrounding model nodes and avoid gaps in the spatial mapping. However, a larger search radius does produce some artifacts around the edges that need to be explained better.
* * *
RC 4) Line 113: I suggest adding some information on the DEM employed in this study.

AC 4) We agree with the reviewer, and added the following information in Section 2.1: "The digital elevation model (DEM) used for this study was generated from a 2015 LiDAR campaign flown over the McMurdo Dry Valleys in 2015. The DEM has 1 m spatial resolution and covers all of Taylor Valley (Fountain et al., 2017)"
* * *
RC 5) Line 134: Here the authors use the -20 C average air temperature of Lake Fryxell

from Obryk et al. (2020) to calculate the age of permafrost. However, this temperature was calculated over a timespan of 30 years, and thus may not be representative of the air temperature since the permafrost refreeze initiation. I understand that the Monte Carlo analysis takes the uncertainty of each parameter into account, but I think there should be a discussion on the reliability of a recent temperature measurement in the context of a much longer time scale.

AC 5) See response to first comment. We have included a variable Tps through time (3C linear cooling trend over the last 10ka). We appreciate the reviewer's comment and believe this will greatly improve the paper.
* * *
RC 6) Line 145: Is a geometric mean appropriate to calculate the bulk thermal conductivity of sediment, fluid, and air mixtures in this scenario? I recommend motivating the usage of a geometric mean over other mixing formulas. For example, Fuchs et al. (2013; https://doi.org/10.1016/j.geothermics.2013.02.002) explore a few different mixing formulas and find that some are better than others for specific sediment mixtures.

AC 6) Even Fuchs et al. (2013) concludes "From the studied models, the geometric mean displays the best, however not satisfying correspondence between calculated and measured BTC" (where BTC stands for Bulk Thermal Conductivity). It is important to note here that the value of BTC for ice-saturated sediments that we calculated using the geometric mean method (2.57 W/m/K in our Table 1) is very close to the BTC (2.55 W/m/K) that one can calculate for the DVDP-6 borehole based on geothermal flux and geothermal gradient given in Table 1 of Morin et al. 2010. The closeness of the two values may be coincidental but it is the only observational point of reference from this region that we can use to calibrate the performance of our BTC calculations. It is also useful to remember that a choice of the exact BTC model is less important in ice-saturated sediments than in water- or air-saturated sediments, for which the existing models have been developed and calibrated (e.g., Fuchs et

al., 2013). This is because the difference between the thermal conductivity of ice and sediment matrix (e.g., 2.2 vs. 2.8 W/m/K in our Table 1) is much smaller than the difference between the thermal conductivity of water and sediment matrix (e.g., 0.57 vs. 2.8 W/m/K). Although we have included the possibility of air saturation in our Monte Carlo model for completeness, the permafrost ages we plot up and discuss in the manuscript are based on assumption of full ice saturation in the bulk of the permafrost layer in our study area. It is our judgement that we cannot substantially improve our model by choosing a different BTC model than the geometric-mean model that we are using right now. Particularly since the calculation of permafrost thickness is only weakly sensitive to BTC of the permafrost layer. For instance, in the analytical solution (Equation 1 in our manuscript), permafrost thickness depends on the square-root of BTC. Hence, even if BTC were to range between 2 and 3 W/m/K, the permafrost thickness would only vary by less than +-10% as compared to the permafrost thicknesses we are now calculating when assuming 2.57 W/m/K.
* * *
RC 7) Line 152: I recommend writing either "variance" or "standard deviation." As it is written in the manuscript, it seems like the two are the same thing.

AC 7) Agreed, we removed the word "variance" from line 152 and line 610, and now only use "standard deviation".
* * *
RC 8) General comment on the results section and related figures: I suggest moving or at least copy some of the text in the result section over to the caption of relevant figures. Currently, the captions are on the minimalistic side, and I believe that adding further explanations would greatly improve the readability of the paper when readers glance through it quickly.

AC 8) Thank you for this suggestion. We have added additional detail to select figure captions to highlight key results of the figures.
* * *
RC 9) Line 247: Inherent -> inherit ?

AC 9) The reviewer is correct. We have replaced inherent with inherit.
* * *
RC 10) Fig. 6 and 7: The usage of this rainbow color scale is problematic for a couple of reasons. (1) It is visually non-linear, with sudden jumps in hue that may result in apparent variability of the dataset that does not actually exist. For example, there is a large jump in light blue-green-yellow that conveniently coincides with the proposed boundary between brine and permafrost resistivities; although this helps locating such putative boundary, I find it potentially misleading. (2) It is very hard to read by colorblind people. To the most kind of color blindness cases, this color scale looks symmetrically identical below and above 200 ohm*m, thus making it very difficult to distinguish which areas are low and high resistivity. Fig. 8 and 13 also employ a non-linear color bar with a large jump mid-range.

AC 10) We understand the concern of the reviewer (particularly with respect to color blind readers). This exact rainbow color scale is very commonly used for airborne geophysics, and is close to being the de facto standard. A linear color would produce images where the structure is visible, but the values on the figure would be completely unreadable. Also a linear scale would be extremely difficult to capture the variation across three orders of magnitude. The log scale balances seeing contrasts in both the low and high resistivity limits (which is needed in this region). We believe it would be far more difficult to find a good looking linear scale that can capture the variations in our resistivity maps that the log scale currently shows well. For these reasons, we propose keeping the color scale as is, but will defer to the editor to make the final call or provide guidance.
* * *
[Figure]

[Figure]

Best fit for Linear Cooling of 3C over 10ka
h = 4.52*t^0.424
R^2 =0.9996

**Fig. 1.**

---

## Author Comment (AC2) · 4 Mar 2021

We want to thank this reviewer for the interest in our paper as well as the very thorough and thoughtful review of our manuscript. It really helped us see the gaps and we were excited to improve it as outlined in our responses below. The lead author especially wants to extend her gratitude, as this is her first time being the lead author on a paper, and this review has been a great learning process.
* * *
My first impression upon reading this paper was, wow, what a novel way to approach this problem! People have been trying to track Dry Valleys lake levels since Scott in

the Heroic Age. None of the existing datasets is complete. Landforms (mostly deltas) dated by radiocarbon tell you when the lake was at a certain level, but geomorphic records are fragmentary and incomplete. Lake bottom sediments from beneath the present lake can give more continuous records, but their interpretation in terms of specific water level is complicated. Here, the authors give a third way of reconstructing former lake levels, which I'm sure has its own associated problems, but which presents a third perspective that may help us circle ever closer to the actual lake history.

That said, I cannot judge the actual methods used in the remote imaging or in interpretation of the resistivity data. It is outside of my field of expertise. I am assuming here that it is correct, and more reliance should be placed on other reviewers on this aspect.

Although my overall impression of the paper is quite favorable, I do think there are some areas that could use reframing and a critical assumption or two that should be revisited.

Reviewer Comment (RC) 1) The analysis seems to operate on the assumption that there was a continuous drop in lake level from the 80 m sill level to the present lake over the second half of the Holocene (and indeed, perhaps from the highstand of GLW). However, this is unlikely to have been the case. This would have been a closed-basin lake and highly sensitive to changes in inputs (almost entirely glacial meltwater) and outputs (evaporation/sublimation). This hydrology places it among a category of lakes that typically shows outsized reactions to small changes in water balance. Pre-Holocene (and even Holocene) lakes in the Dry Valleys region are suggested to have undergone large-scale fluctuations during the last glaciation and termination of the ice age (i.e., Hall et al., 2010, PNAS - a reference which probably should be included). Lake Vanda showed large fluctuations at 3 and 6 ka. So, it is likely that Lake Fryxell did as well - and possibly other changes not yet documented. To what degree does this lake level volatility affect your modelling results and error bars? How long must permafrost be covered up to completely melt under a lake (I assume this results from permafrost thickness and length of lake cover)? In short, it seems as if this assumption

may be quite critical to the outcome of the model.

Author Comment (AC) 1) We agree that multiple lake level fluctuations in both the filling and draining directions could have occurred throughout the Holocene. Our results do not necessarily assume continuous lake level drop - it would be the same result if the lake level drop was instantaneous. We have made this clearer in the text.

When it comes to analyzing the potential importance of putative lake level variability at elevations <81 m.a.s.l., it is important to note in our Figure 9 that the cluster of thin permafrost at low elevations occurs for elevations <50 m.a.s.l. where permafrost is estimated to be <50 m thick. Comparatively few of our estimates of permafrost thickness fall between 50 and 140 m with corresponding surface elevations between 50 and 81 m.a.s.l. The only other high-abundance cluster of permafrost thickness is 140-200 m with corresponding surface elevations between 81 and 150 m.a.s.l. This cluster represents the lake levels that occurred when the ice dam was still present. When the ice dam is not present, the hydrology of Fryxell basin favors lake levels that are considerably lower than the sill level of 81 m.a.s.l. and that reach mostly up to ca. 50 m.a.s.l., which corresponds to the other cluster of common permafrost thicknesses that are <50m thick. We created a model to explore this effect (fill drain cycles), however we are not including this in the paper because we do not believe we can resolve any individual lake level fluctuation events. Our modeling indicated that permafrost that is <50 m thin takes only ca. 150-300 years to grow, because permafrost growth rates are initially high. Thin permafrost is also most susceptible to melting away relatively fast when submerged by rising lake levels. Hence, it is not likely that our conclusion that the cluster of thin permafrost is very young can be substantially changed by assuming fluctuation of lake levels.
* * *
RC 2) The paper attempts to compare model results with 14C and OSL data. However, this is mixing of apples and oranges (and may stem from the assumption in point 1 above). Both the 14C and OSL dates refer to the position of the lake at a specific time.

They do not preclude the lake from reaching that same level (or higher) at another, later time. It is highly likely that most if not all of the dated deltas were under water at multiple times, not necessarily reflected by the dates (this is to some degree acknowledged in the discussion of OSL, but the same applies to radiocarbon). Some deltas at critical levels (i.e., the sill level) may have been occupied and active at several times. Thus, in my opinion, you cannot make a direct comparison of the three different dating methods, simply on elevation. These lakes fluctuated many times and there are going to be deposits from earlier times mixed in at the same elevations occupied by later lakes. Recognition/resolution of this assumption may help resolve some other issues raised below.

AC 2) We agree totally that 14C, OSL and our model results are measuring different things. But an important point to make is that we are not presenting our permafrost age model as a precise dating technique. It is merely showing that the most recent occupation of the area by a large lake happened on a time scale more consistent with the younger OSL chronology than the older 14C chronology. We also agree that there would have been fluctuating lake levels during the Holocene. But having said that, we note that the OSL data from Berger et al. (2013), if taken at face value, does suggest a fairly smooth curve of lake level decline over the last ∼13k years, but this is hard to say conclusively without more OSL data. The 14C is indeed, all over the place, but does show a general trend of lake level lowering over time. Likewise, the uncertainty of our model results grows large with time.

We understand the reviewer's concern and feel that since we can't resolve the errors in the various approaches, the best thing to do is make sure there is appropriate discussion of the differences in the different data types, and that we are making broad comparisons. We have modified Figure 14 (which compares all three methods), and instead of shading regions with raw data, 1 sigma, and median line, we have displayed the data similar to Figure 9 as individual model points. We still plan to include the OSL and 14C data for comparison. We have rewritten the paragraph starting on line 236

with this goal. Here it is:

"With inputs tuned to maximize permafrost ages, it was still not possible to yield ages for lake occupation comparable to those estimated by 14C ages of delta deposits (Hall and Denton, 2000) (Fig. 14). Radiocarbon dating from Hall and Denton (2000) suggests that lake levels above modern levels occurred between 22 to 5 ka BP, and there are no 14C samples to suggest that levels exceed modern lake elevation after 5 ka BP. OSL dates (Berger et al., 2013) estimate past lake level high stands existed between 12 – 5 ka BP, and do not suggest recent lake level high stands. Our permafrost age calculations agree more with the OSL chronology than the 14C (Fig. 14), both have limitations. OSL data records the burial age of the quartz grain, which may be different from the deposition age. Rates of past sediment transport and deposition are not currently known, so this lag in deposition versus burial time is unresolved. Secondly, OSL dates are collected from depth, meaning the dates may not be an accurate representation of the most recent occupation of that delta/lake level elevation (this will also be true for 14C samples collected away from the surface). Several studies have shown a substantial radiocarbon reservoir effect in the MDVs, but modern lake edges and streams have been shown to be mostly-well equilibrated with the modern atmosphere (e.g. Doran et al. 1999; Hendy and Hall 2006). Doran et al, 2014 did note some exceptions. In fact, out of 4 moat microbial mat samples dated (all in < 1m of water), only one was equilibrated with the modern atmosphere. The others carried 14C ages of 2324±96, 9334±71, 2608±48. So clearly, even in modern lakes it is possible for lake edge material to carry a significant carbon reservoir. A large glacial lake of the past may have had even more ancient unequilibrated carbon associated with it due to melt coming from the ancient RIS without the opportunity for significant equalization with the atmosphere (e.g. through direct subaqueous glacial melt and moats being more closed to the atmosphere)."

———————————————————————————————————————————————

RC 3) I am having some difficulty with Fig. 14. If I am understanding it correctly, parts of

the modelled curve become problematic and require some explanation. I have already mentioned the 14C and OSL data - I don't think you can compare them directly for reasons mentioned above and recommend taking them off this diagram. My concern here is about the model results themselves.

My understanding is that this figure shows the mean of the model (solid line), a dark shaded zone (1 sigma), and a light shaded zone (raw data). The underlying assumption here is that the permafrost age is directly linked to the lake presence (let's revisit this in a minute). Thus, the older permafrost ages at higher elevations are attributable to the lake dropping from there earlier. Those at >80 m must be from a time when there was still an ice dam at the valley mouth, because they are higher than the sill. However, model ages for >80 m seem far too young. Measurements by multiple methods place deglaciation at the mouth of Taylor Valley by 8 ka if not a bit earlier. On Fig. 14, 8 ka intersects the model results (solid line) at 160 m elevation, something that is not possible, because there would have been no ice to dam the lake. Even the 1 sigma error bars are well above the 80 m sill. The lake cannot exist above 80 m after 8 ka, so how is this result explained? Only the low end of the raw data fit in the plausible zone (>80 m = >8ka). Is there something about the model or in the picking of 100 m that is producing results skewed toward young ages? How does this affect the reliability of the conclusions in this paper? In other words, if we know the model is producing erroneous ages above 80 m, why should we have confidence in results below 80 m elevation? Greater discussion would be useful here.

AC 3) We appreciate this comment from the reviewer, and agree that this topic deserves more discussion. It is correct that the assumption that permafrost age is directly linked to the presence of a lake. However, the exact date that the ice dam is removed is still not completely resolved in the literature. Based on OSL, Berger et al. (2013) suggests a lake level drop below sill level between 5 to 8ka. Hall and Denton (2000) reports one 14C sample from Explorer's Cove that suggests Ross Ice Sheet was last grounded in Explorer's Cove ~6ka BP. Additionally, Levy et al. (2017) uses luminescence dating to suggest that large paleolakes in neighboring Garwood Valley persisted well after the Ross Ice Sheet retreated. The paleolake Howard (Garwood Valley) was at its maximum elevation until 4.26 +/- 0.72 ka because of stranded ice in the mouth of Garwood Valley that was a relic of the Ross Ice Sheet. Levy et al. (2017) makes the case that relic ice could have persisted in the mouths of various valleys well after the Ross Ice Sheet retreated in McMurdo Sound. Levy et al. (2017) points out that radiocarbon samples (14C) have consistently produced dates 5 - 10 ka older than OSL samples from the same locations, so we think our paper is within this argument that 14C likely overestimates paleolake age by thousands of years. For this reason, we think it is reasonable, based on the literature, to consider younger (than 8 ka) ages for a lake above 81 m asl.

Because of this reviewer's comment as well as another reviewer's comments, we have decided to include a second approach to calculating permafrost ages using a 1D numerical (finite-difference) model solving the classical Stefan problem of vertical heat diffusion coupled with latent heat release during freezing (see attached Figure A). The upper boundary condition is a prescribed temperature that is deltaT lower than the freezing point of the sub-permafrost brines (deltaT = surface temperature - brine freezing temperature). This deltaT can be either held constant during numerical experiments or can be prescribed to vary with time. The bottom boundary condition of the numerical model is a constant heat flux, set to the geothermal flux of 0.080 W/m2 consistent with two borehole-based estimates proximal to the study area (boreholes DVDP-6 and CIROS-1 in table 1, Morin et al., 2010). Other model parameters are based on permafrost properties listed in Table 1 in our original manuscript.

Existing observational constraints indicate that under modern conditions in the study area the temperature at the bottom of the permafrost is ca. -9C (e.g., Figure 7 in Foley et al., 2015) while ground surface temperature is ca. -19C (Table 1 in Obryk et al., 2020), yielding deltaT of about 10C. When we assume a constant deltaT of 10C, our numerical experiment still yields fairly young ages for elevations below the 81 masl sill

level (<4ka). We then applied a linear cooling rate of 1, 2, 3, and 4C over the last 10,000 yr to model the cooling trend observed in the Holocene Taylor Dome paleotemperature reconstructions (Stieg et al., 2000, Monnin et al., 2004). These ice core constraints are best approximated by a linear cooling trend of 3C per 10,000 years. In the revised version of the manuscript we have included these results to create an additional figure (similar to Figure 13) which shows the permafrost age distribution using the 1D vertical diffusion model.

This method still yields fairly young ages for elevations below the 81 masl sill level (<150 m thick permafrost results in ∼4ka, green line below). This would be about 2.5 ka older than our original approach using Eq 1 (Osterkamp and Burns, 2003). Even though the results vary between the two methods, the data still suggests that a lake occupied Fryxell basin between 1.5ka (analytical method) to 4ka (1D vertical diffusion model explained above). Higher elevations (permafrost thicknesses >/=200 m) yields ages between 7 - 8ka.

This paper discusses Glacial Lake Washburn in the context of past lake level highstands, however we are not trying to necessarily date the timing of the highest lake levels. The errors in our approach are greatest where permafrost is thickest (and hence oldest), so other methods to determine very old lake levels are likely better than our method of using resistivity data. We are simply suggesting that lake levels have been at sill level within the last 5ka.
* * *
RC 4) While it seems plausible, is changing lake level the only variable that can affect permafrost age? How old is that brine?

AC 4) An extensive body of scientific work documents that surface climatic and hydrologic conditions in Taylor Valley have varied considerably on timescales ranging from decades to millennia (e.g., Hall et al., 2000; Doran et al., 2002, etc.) Much less is known about subsurface variability in the study region. However, it is generally

assumed that on Earth the subsurface is hydrologically and thermally less dynamic than subsurface. The diffusive nature of heat transfer near the surface of the Earth rapidly mutes the amplitude of surface climate fluctuations as depth below the surface increases. For instance, Clow and Waddington (1996) reported a re-measurement of the vertical temperature profile in the DVDP-11 borehole, which is contained within our study area. They observed that 1degreeC of ground surface warming between the 1970s and 1990s decade to nil within ca. 70 meters from the surface. We had the opportunity to re-measure subsurface resistivity in the lower part of Taylor valley between 2011 and 2018. Our comparison of the two datasets does not reveal any discernible subsurface resistivity changes. We also used the regional measurements of resistivity to infer that the subsurface groundwater system in Taylor Valley and neighboring valleys is quite sluggish (Foley et al., 2019). Our upper-end estimate for groundwater velocity between Lake Fryxell and the mouth of the Taylor Valley is 0.01 m/yr. This means that even within the longest timescales considered in our manuscript (ca. 10,000 years), groundwater may have moved horizontally by a total of <100 meters. The only other process we can think of that may, in principle, contribute to temporal variability of properties in the sub-permafrost brine is molecular diffusion of ions in pore spaces. However, the diffusivity coefficients for ions like Na+, Cl-, and other major constituents of Taylor Valley brines is orders of magnitude lower than thermal diffusivity so this process should be even more sluggish than the diffusion of surface temperature changes that we considered above. Therefore, we expect the properties of the sub-permafrost brine to be relatively constant on timescales of centuries and millennia considered in our manuscript. When it comes to the related question of the age of these brines, there are no direct estimates for the subsurface brines in Lower Taylor Valley because these brines have never been sampled. However, the brines in Lake Bonney and at Blood Falls have been studied in more detail and most recent estimates of their age invoke ages of the order of hundreds of thousands of years to millions of years (e.g., Mikucki et a., 2009; 2015; Foley et al., 2019; Blackburn 2020). These brines were formed through cryoconcentration of seawater and the last time

that seawater may have been emplaced into the subsurface of this coastal region is in the Pliocene or Miocene. This is one reason why our model of the subsurface brine system in Taylor Valley envisions the brines to be old and slowly evolving.
* * *
Minor Points (by line)

RC 5) Line 49 - use the calibrated value, not the raw 14C age.

AC5) We have changed the age on Line 49 from the calibrated value for 12,700 14C yr BP to 13,067 yr BP using the Calib Program. (Stuiver et al., 2017)
* * *
RC 6) Line 54 - The Taylor Dome Holocene chronology has been revised (Monnin et al., 2004), which actually is fairly substantial in the mid-Holocene. Use of any specific times should be checked as, if my memory is right, parts of the chronology have shifted as much as 1500 years.

AC 6) Thanks for catching this. We agree with the reviewer's suggestion to use the latest Taylor Dome chronology, and we replaced Figure 2 with a modified figure from Monnin et al. (2004). We have used Figure 4D from Monnin et al. (2004), and add a left y axis showing temperature deviation from modern. We used the D18O to delta T conversion shown in our original Figure 2 from Steig et al. (2000).
* * *
RC 7) Line 61 - "atmospheric correction" is an odd way to put it. I would simply call them calibrated ages. Hall et al., 2010 in PNAS is a key paper here with an updated dataset (see the SI with that paper).

AC 7) We agree with the reviewer and the sentence now reads "The calibrated 14C ages from Hall and Denton (2000) shown in Figure 4 were corrected using the CALIB program (Stuiver et al., 2018)."
* * *
RC 8) Line 67 - I'm not sure the lack of samples led to an assumption that lakes remained at or below modern levels over the last 5000 years but only that there were no data and that Lake Fryxell had to be below the sill.

AC 8) We agree that the lack of younger than 5k yr deltas is suggestive, not conclusive, and have removed this sentence. ___________________________________________________________________

RC 9) Line 106 - says <1000 m but line 115 and elsewhere uses 100 m as the cutoff. Is this a typo or something else? If not a typo, why was 100 chosen?

AC 9) We chose 100 ohm-m as our boundary between permafrost and brine, so this wasn't a typo. However, the reviewer's comment shows that we need to clarify why we chose 100 ohm-m, and why we mentioned 1000 ohm-m as another threshold.

Permafrost can be broken into subgroups, depending on degree of saturation and confining properties as described in McGinnis and Jensen (1971). Confining permafrost does not allow any fluid flow, whereas aquifrost is permafrost that allows groundwater flow due to local conditions such as salinity and porosity (McGinnis and Jensen, 1971). Confining layer permafrost tends to have much higher electrical resistivities (<10,000 $\Omega$m) than aquifrosts (50 - 1,000 $\Omega$m) depending on temperature and degree of saturation of the aquifrost (McGinnis and Jensen, 1971).

The 1000 ohm-m threshold comes from both McGinnis and Jensen (1971) as well as the Mikucki et al. (2015) paper which broadly defines sediments as brine-bearing if they have resistivity from 10 to 1000 ohm-m. We are using the term "high" versus "low" resistivity to broadly explain how the depth of investigation varies.

The 100 ohm-m cut off was chosen using Foley et al. (2015, Figure 7) Dry Valley Drilling Project (DVDP) borehole data comparison that shows resistivity measurements. The borehole DVDP-11 was drilled within our study area and had temperature and salinity measured. These borehole data can be compared to our resistivity profile taken near the borehole site (Foley et al., 2015, Figure 7). Their figure shows a rapid decline in resistivity from about 100 ohm-m down to down to <5 ohm-m within only 20-30 m. This sharp resistivity gradient corresponds well to an abrupt jump in salinity with depth. Such a salinity increase is consistent with a transition from frozen sediments to liquid brine. This is how we justify our choice of 100 ohm-m as the nominal boundary between permafrost and liquid brine. If we were to choose a threshold value lower than that, the permafrost thicknesses we would get would be marginally greater (ca. 20-30 m). If we used a higher cut off (say 500 ohm-m), the permafrost layers would be even thinner, and therefore yield even younger ages.
* * *
RC 10) Line 196 - see comment above about these deltas being covered and recovered by water and thus direct comparison is impossible.

AC 10) This will be covered by a new paragraph in the Discussion, and we have added words of warning of direct comparison throughout.
* * *
RC 11) Line 205 - this distance would be made more useful for lake level if an elevation were associated with it.

AC 11) We used a distance from lake edge for this case because the ground-water system is assumed to be confined by permafrost from above and bedrock below. The groundwater system extends laterally from the lake and has been freezing back after lake levels dropped. Instead of changing the text, we added a reference to Figure 7 at the end of the sentence on Line 205.
* * *
RC 12) Line 213 - this assumption is critical and might be key.

AC 12) It is not clear to us that this assumption is as consequential to our results and their discussion as the reviewer presupposes. One of the most important issues

treated in our paper is the question of when the Fryxell basin was filled up with water to above the sill level of ca. 81 m.a.s.l. Such a deep lake required an ice dam. We are focused on using the estimated permafrost thickness to estimate when the ice dam was last present at the mouth of Taylor Valley. Once the ice dam was gone, it did not come back so lake levels could not reach above 81 m.a.s.l anymore. Hence, the issue of lake-level cycling is not really relevant to our analyses of the permafrost with surface elevation >81 m.a.s.l. One could argue that maybe paleo-lake levels varied above 81 m.a.s.l. before the final disappearance of the ice dam. This could result in some of the permafrost thickness observed at these elevations being formed already before the final disappearance of the ice dam. This, in turn, would push our estimate of the final ice dam disappearance to be younger than it already is. See above comment (response to Major Point 1) for further explanation about how < 81 masl (below sill level) fluctuations would impact permafrost age.

In order to better address and highlight these assumptions, we have provided further discussion about how Tps and lake level fluctuation would affect the resistivity signal. Another reviewer also pointed this out as needing further explanation.

Regarding constant Tps assumption, as mentioned above, we have included a second 1D model that allows us to prescribe a variable Tps through time (we chose a linear cooling rate of 3C over 10000 years). This yields slightly older ages.

Regarding the constant lake level drop assumption - we will clarify that this analytical approach does not distinguish between constant drop or instantaneous drop, but either way it does not take into account multiple cycles of fill/drop that could have definitely occurred. This is especially true for recent lake level history without an ice dam (<81 masl). As mentioned above, we applied a cyclical fil/drain cycle to the 1D model. We have not included this figure in the paper, but have provided more explanation about what various fill/drain cycles would mean for our interpretations.

RC 13) Line 225 - this paragraph goes with the assumption of gradually lowering lake, which was not the case. See Hall et al., 2010. It also makes the assumption of a close link between air temperature at Taylor Dome and lake level, which may be problematic. Also, see comment above about revised Taylor Dome Holocene timescale.

AC 13) Our results do not necessarily assume continuous lake level drop - it would be the same result if the lake level drop was instantaneous. See comments above. We make sure to clarify this in the text.

We will rewrite this paragraph starting on Line 225 to remove the definitive language talking about exact ice dam removal, because as this reviewer pointed out, our results show younger permafrost ages above the sill level.

Regarding the revised Taylor Dome Holocene timescale and associated temperatures, we mentioned above that we will replace the Steig et al. (2000) Holocene temperature (Figure 2) with a modified figure from Monnin et al. (2004). We understand the reviewer's comment about the link between air temperature at Taylor Dome and lake level, and will give this topic more discussion. We recognize that there is more that goes into lake level change than simply air temperature alone: changes in albedo (snow on landscape) and incoming radiation will change the energy balance, and subsequently the stream/lake response, but you have to have temperatures near freezing for melt to happen. These limitations will also be included in order to clarify the dynamic relationship between lake level and meteorological conditions.
* * *
RC 14) Line 238 - I may have missed it, but I didn't see anywhere in Hall and Denton (2000) where they stated that lake levels had remained at or below present since 5 ka. They didn't have data to address this.

AC 14) We agree and have clarified this assumption throughout the paper (mentioned above in previous comment as well). Specific to Line 238, we rewrote it saying "Radiocarbon dating from Hall and Denton (2000) suggests that lake levels

were higher than modern levels between 22 to 5 ka BP, however Hall and Denton (2000) do not provide evidence of lake levels exceeding modern levels after 5ka."
* * *
RC 15) Line 243 - the sentence about OSL dates from depth and not applying to most recent episode of lake level change applies equally to radiocarbon.

AC 15) We agree and have made changes in this section of the paper to reflect this.
* * *
RC 16) Line 249 - application of a large reservoir correction to shallow water lake sediments seems ad hoc, given prior references. Most of these deltas formed at the same stream mouths as today, fed locally by the same glaciers as today, not the RIS. As the data should not be compared directly (for reasons already given), such an attempt at correction is not warranted.

AC 16) See comments on this above. The data we are comparing in this section of the paper is 14C and OSL. The argument we are making is that in a large glacial lake, there may have been more pathways for ancient carbon to remain unequilibrated with the atmosphere - for instance - at the glacier edge there would have been a large subaqueous ice wall melting directly into the lake without contact with the atmosphere. There would have been large volumes of supraglacial melt in contact with and eroding ancient ice until it entered the lake. The moats may have been much less open than today. None of this seems unreasonable, especially when considering moat microbial mat ages in the modern lakes shown by Doran et al 2013 between 2k and 6k 14C yrs. The preserved mat material in the ancient deltas is from these lake edge mats the stream bed is moving over, not the stream beds themselves.
* * *
RC 17) Line 259 Whittaker here and later in paragraph.

AC 17) Thank you for pointing out this missing reference. We have added

the following reference. Whittaker, T., Hall, B., Hendy, C., and Spaulding, S. Holocene depositional environments and surface-level changes at Lake Fryxell, Antarctica. The Holocene, Volume 18, Issue 5, Pages 775-786 (2008)
* * *
RC 18) Par. starting with 265. While I don't necessarily disagree with the conclusion of the last drop in lake level being post 1.5 ka, I find these stated reasons unconvincing. The link with the Taylor Dome ice core here implies, without specifically stating, that high temperatures should lead to high lake levels. But, this may not be the case. If you want to make this argument, this needs to be stated explicitly (and you'll want to confirm that the chronology didn't shift too much for this to still apply). In addition, there are many, large, well-preserved deltas and while the Crescent Stream one is a nice one, weathering rates are so low that you cannot resolve 1000 yrs vs 10,000 years based on delta appearance.

AC 18) We switched to Monnin et al. (2004) timescales per the recommendation of this reviewer, and their reconstruction suggests that temperatures were above modern from ∼1,800 to 12,000 yr BP. This new record stops at 1ka unfortunately.

Similar to the reviewer's comment about Line 225. We have taken the reviewer's suggestion of explicitly stating the relationship between temperature and lake level. We will support this claim using the following references: Obryk et al. (2017), shows how warming due to foehn winds could be the cause of higher lake levels during the LGM. Doran et al. (2008) which shows how a summer with record single season lake level change was connected to temperature (degree days above freezing). And Doran et al. (2001) which shows a decadal lake level decrease due to cooling. Like mentioned above in the response to Line 225 comment, we recognize that other meteorological variables impact lake level and have highlighted this.

We are not trying to date the delta from its appearance, but simply pointing out that the best preserved delta in the entire Fryxell basin is at the

sill elevation. We have included a sentence to further highlight that we cannot quantify the relationship between delta appearance with delta age due to many unknown variables (age, deposition rates, erosion rates, etc).
* * *
RC 19) Be careful with the 6 degrees warmer comments about Taylor Dome. Temperature is only one of several variables that affect stable isotope data. In addition, the 100 yr running average in the Steig paper (probably more applicable to lake-level changes than any single year) suggests only about 1 degree change and is probably a better representation of actual temperature changes.

I think these reasons don't help your argument any. Why not just say your data suggest the lake might have last been at the level of the sill at 1.5-1 ka and leave it at that? To my knowledge, there is nothing that says it couldn't have been. The OSL and 14C data are older, but they may be dating earlier events.

AC 19) We agree that the 6C should be removed, especially because we took the reviewer's suggestion to use Monnin et al. (2004) updated timeline. We will simply point out that temperatures were warmer than modern during the Holocene.
* * *
RC 20) Line 287 and elsewhere. GLW refers only to the ice-dammed lake. Once the ice sheet retreated, this became Lake Fryxell.

AC 20) This is correct, and we will be sure to only use the term Glacial Lake Washburn (GLW) when referring to the paleolake when lake levels were higher than the sill level due to grounded ice in the mouth of Taylor Valley. We will ensure this change is made throughout the manuscript.
* * *
RC 21) Please double check references. I didn't check them all, but the one I looked up to make sure it was spelled correctly (Whittaker) wasn't there.
AC 21) We will check all references, and as mentioned above, will add Whittaker et al. (2008) to the list. Thank you!
* * *
Fig 1) Please add latitude/longitude

We added Latitude and Longitude to Figure 1.
* * *
Fig 2) You may want to switch to the Monnin et al., 2004 timescale.

We agree with the reviewers' suggestion to use the latest Taylor Dome chronology, and we will replace Figure 2 with a modified figure from Monnin et al., 2004. We plan to use Figure 4D from Monnin et al., 2004, and add a left y axis showing temperature deviation from modern. We will use the D18O to delta T conversion shown in our original Figure 2 from Steig et al., 2000.
* * *
Fig 3) May want to mark the location of this figure on Fig. 1

We added a polygon to show the outline of Taylor Valley on Figure 1.
* * *
Fig 4) suggest changing "atm corrected" to "calibrated"

We agree, and we changed the legend to read "calibrated 14C".
* * *
Fig 6) I found the color on these maps (this and the ones that follow) hard to follow, I'm afraid. Is there a better way to present this? The caption could be more informative for those (like me) who do not deal in resistivity commonly. Perhaps you could add a sentence about what is meant by constant elevation.

This exact rainbow color scale is very commonly used for airborne geophysics, and is close to being the de facto standard. A linear color would produce images where the structure is visible, but the values on the figure would be completely unreadable. Also a linear scale would be extremely difficult to capture the variation across three orders of

magnitude. The log scale balances seeing contrasts in both the low and high resistivity limits (which is needed in this region). We believe it would be far more difficult to find a good looking linear scale that can capture the variations in our resistivity maps that the log scale currently shows well. For these reasons, we propose keeping the color scale as is, but will defer to the editor to make the final call or provide guidance.

Thank you for this suggestion about the caption, and we will include the following sentence. "Constant elevation refers to a horizontal slice through the subsurface at depth (in this case, 100 m below sea level) that is 5m thick." We may also include a light grey shading of this elevation and thickness on Figure 7 to help readers connect the two figures.
* * *
Fig 13) see comment above about using GLW. This should be "lowering of Lake Fryxell" rather than draining of GLW, which would have happened when the ice sheet retreated. Draining really isn't the right word for something that evaporated.

Thank you for pointing this out and we agree it is an important distinction. We will pay special attention to the usage of the term Glacial Lake Washburn, and reserve it for lake levels higher than the sill level (81 masl).
* * *
Fig 14) was covered in comments above. Suggest removing other chronologic data as the comparison doesn't really mean anything and is potentially misleading.

We addressed the comments about apples and oranges elsewhere. We feel we have provided enough context, caveats and explanation of the various approaches and what "age" means (they all report age) that this comparison is appropriate. The main takeaway is there is some discrepancy between the 14C and OSL chronology, and the permafrost modeling agrees more closely with the latter. It is difficult to reconcile the permafrost ages with the older 14C chronology, no matter how we turn the knobs of the model. We are not trying to establish permafrost age modeling as a dating technique, but in this case it acts as an independent check on the two chronologies.

Overall Impression Although I have pointed out several potential issues with the paper, I think the approach is really intriguing and the study is important to publish. I think most of these comments can be dealt with fairly quickly. Assessing the impact of the assumption about constant lake-level drop (which is critical), may take more effort. I recommend publication after moderate revision.

[Figure]

Fig. 1.

[Figure]

---

## Editor Decision (ED1)

Dear Ms. Myers and co-authors,

Thank you for the submission of your response to the Reviewers' comments. I am pleased to see a positive set of reviews, with some constructive suggestions that you have addressed in your initial response. The revised Figure 6 is a significant improvement. I am happy to accept your paper for publication in the journal, subject to the minor technical corrections listed below.

Reviewer Nerozzi assessed the revision and makes the following comments on the revised manuscript:

Fig. 5: I'm personally against connecting dots in a plot unless there is a good reason to do so. This is because that assumes a linear and constant change between two measurements, which is likely not the case give the high interannual lake level variability. However, I'll leave the choice of keeping the lines or not to the authors.
- Fig. 6: The choice of colors for resistivity is definitely better than the previous version of the same figure. However, my recommendation is still to opt for a more linear scale - the jump is resistivity between brine and permafrost is dramatic (100 to 1000 ohms over a very short distance) and I bet it would still show up even with a very simple visually linear scale.

Therefore I ask you to consider these points carefully before uploading your final revised manuscript. I know you have invested significant time into redrawing the figures (especially Fig. 6). If they can be modified again without too much difficulty I would suggest following the Reviewer's suggestions to increase the accessibility of your paper to the widest possible audience.

Please do let me know if you think this is possible within a reasonable timescale, or whether you would prefer to keep the figures as they are. I now request that you upload your updated manuscript including the changes detailed in your response.

Thank you for your contribution to The Cryosphere, I look forward to reading your revised manuscript.

Kind regards,

Dr Liz Bagshaw

---

## Author Response (AR2)

"Thermal legacy of a large paleolake in Taylor Valley, East Antarctica as evidenced by an airborne electromagnetic survey"
by Krista F. Myers et al.

Response to Reviewer Report #1, Steano Nerozzi & Editor, Dr. Liz Bagshaw
Initial response from authors: 4 June 2021
Final response from authors: 24 June 2021

**Editor Comment:**

Dear Ms. Myers and co-authors,

Thank you for the submission of your response to the Reviewers' comments. I am pleased to see a positive set of reviews, with some constructive suggestions that you have addressed in your initial response. The revised Figure 6 is a significant improvement. I am happy to accept your paper for publication in the journal, subject to the minor technical corrections listed below.

Reviewer Nerozzi assessed the revision and makes the following comments on the revised manuscript:

Fig. 5: I'm personally against connecting dots in a plot unless there is a good reason to do so. This is because that assumes a linear and constant change between two measurements, which is likely not the case give the high interannual lake level variability. However, I'll leave the choice of keeping the lines or not to the authors.

> **Author Response: We appreciate this comment, and we have removed the lines from the lake level plot (Figure 5) because we agree that it is incorrect to assume a linear and constant change between two measurements of lake level.**

- Fig. 6: The choice of colors for resistivity is definitely better than the previous version of the same figure. However, my recommendation is still to opt for a more linear scale - the jump is resistivity between brine and permafrost is dramatic (100

to 1000 ohms over a very short distance) and I bet it would still show up even with a very simple visually linear scale.

Therefore I ask you to consider these points carefully before uploading your final revised manuscript. I know you have invested significant time into redrawing the figures (especially Fig. 6). If they can be modified again without too much difficulty I would suggest following the Reviewer's suggestions to increase the accessibility of your paper to the widest possible audience.

**Author Response: We are glad to hear that the reviewer and editor agree that the color ramp is an improvement, however we are not going to revise the figure with a linear color ramp because a) A linear color would produce images where the structure is visible, but the values on the figure would be completely unreadable. Also a linear scale would be extremely difficult to capture the variation across three orders of magnitude. The log scale balances seeing contrasts in both the low and high resistivity limits (which is needed in this region), and b) If we were to make this edit, it would require re-doing the figures in the specialized inversion software (Workbench), which would take a lot of time. We appreciate that the reviewer and editor acknowledge that this type of modification is not quick, which is why we are not going to make this revision.**

Please do let me know if you think this is possible within a reasonable timescale, or whether you would prefer to keep the figures as they are. I now request that you upload your updated manuscript including the changes detailed in your response. Thank you for your contribution to The Cryosphere, I look forward to reading your revised manuscript.

Kind regards,
Dr Liz Bagshaw